# Ribonuclease activity undermines immune sensing of naked extracellular RNA

## Graphical abstract

## Highlights

- Naked extracellular RNA (exRNA) is bioactive when RNases are inhibited/absent

- Internalized naked exRNA activates endosomal TLRs as well as cytosolic RLRs

- Naked extracellular mRNAs can be translated in recipient cells

- Extracellular RNases avoid naked-exRNA-induced systemic inflammation

## Authors

Mauricio Castellano, Valentina Blanco, Marco Li Calzi, ..., Alfonso Cayota, Mercedes Segovia, Juan Pablo Tosar

## Correspondence

msegovia@pasteur.edu.uy (M.S.), jptosar@pasteur.edu.uy (J.P.T.)

## In brief

Castellano et al. show that when extracellular RNases are inhibited or absent, naked extracellular RNA is internalized spontaneously by cells, inducing inflammation mediated by endosomal TLRs or cytosolic RLRs. Extracellular ribonucleases counteract this effect. The authors also show translation of naked extracellular mRNAs upon ribonuclease inhibition.

 Castellano et al., 2025, Cell Genomics 5, 100874
May 14, 2025 © 2025 The Authors. Published by Elsevier Inc.

CellPress

Article

# Ribonuclease activity undermines immune sensing of naked extracellular RNA

Mauricio Castellano,[1,2,3] Valentina Blanco,[1] Marco Li Calzi,[1] Bruno Costa,[1,3] Kenneth Witwer,[4,5,6] Marcelo Hill,[2,7] Alfonso Cayota,[1,8] Mercedes Segovia,[2,7,*] and Juan Pablo Tosar[1,3,9,*]

[1]Functional Genomics Laboratory, Institut Pasteur de Montevideo, Montevideo 11400, Uruguay
[2]Immunoregulation and Inflammation Laboratory, Institut Pasteur de Montevideo, Montevideo 11400, Uruguay
[3]Analytical Biochemistry Unit, School of Science, Universidad de la República, Montevideo 11400, Uruguay
[4]Department of Molecular and Comparative Pathobiology, Johns Hopkins University School of Medicine, Baltimore, MD 21205, USA
[5]EV Core Facility "EXCEL," Institute for Basic Biomedical Sciences, Johns Hopkins University School of Medicine, Baltimore, MD 21205, USA
[6]The Richman Family Precision Medicine Center of Excellence in Alzheimer's Disease, Johns Hopkins University School of Medicine, Baltimore, MD 21205, USA
[7]Academic Unit of Immunobiology, School of Medicine, Universidad de la República, Montevideo 11800, Uruguay
[8]Hospital de Clínicas, Universidad de la República, Montevideo 11600, Uruguay
[9]Lead contact
*Correspondence: msegovia@pasteur.edu.uy (M.S.), jptosar@pasteur.edu.uy (J.P.T.)

## SUMMARY

Cell membranes are thought of as barriers to extracellular RNA (exRNA) uptake. While naked exRNAs can be spontaneously internalized by certain cells, functional cytosolic delivery has been rarely observed. Here, we show that extracellular ribonucleases (RNases)—primarily from cell culture supplements—have obscured the study of exRNA functionality. When ribonuclease inhibitor (RI) is added to cell cultures, naked exRNAs can trigger pro-inflammatory responses in dendritic cells and macrophages, largely via endosomal Toll-like receptors (TLRs). Moreover, naked exRNAs can escape endosomes, engaging cytosolic RNA sensors. In addition, naked extracellular mRNAs can be spontaneously internalized and translated by various cell types in an RI-dependent manner. *In vivo*, RI co-injection amplifies naked-RNA-induced activation of splenic lymphocytes and myeloid leukocytes. Furthermore, naked RNA is inherently pro-inflammatory in RNase-poor compartments like the peritoneal cavity. These findings demonstrate that naked RNA is bioactive without requiring vesicular encapsulation, making a case for nonvesicular-exRNA-mediated intercellular communication.

## INTRODUCTION

Lipid bilayers are considered to be impermeable to negatively charged RNA molecules. However, cells can pack and release RNA inside biological lipid nanoparticles called extracellular vesicles (EVs).[1–3] These vesicles can be internalized by recipient cells enabling the intercellular exchange of genetic information.[4,5] Similarly, therapeutic delivery of messenger RNA (mRNA) or double-stranded small interfering RNA (siRNA) also requires RNA encapsulation inside synthetic lipid nanoparticles (LNPs).

The spontaneous uptake of naked RNA molecules is less studied and considered highly inefficient. Nevertheless, some pharmaceutical preparations containing antisense oligonucleotides (ASOs) lack cationic lipids or any other transfection reagents.[6] These drugs manage to enter cells and gain access to the cytosol or even the nucleus through an endocytosis-dependent naked-RNA-specific uptake process referred to as gymnosis.[7] Although the molecular mechanism(s) responsible for this functional uptake route are still obscure, gymnosis might depend on the increased hydrophobicity conferred by phosphorothioate bonds, a frequent modification in therapeutic ASOs.[8]

Gymnotic uptake and subsequent translation of naked mRNA vaccines has been reported, usually following intramuscular, intradermal, or intralymphatic administration in mice.[9–12] These studies have also shown that dendritic cells are the main cell type responsible for gymnotic mRNA uptake, probably through a macropinocytosis-dependent mechanism. However, intranodal injection of naked mRNA vaccines in humans failed to show a response compared with placebo.[13] The clinical success of LNPs as delivery vectors for mRNAs and siRNAs[14,15] has consolidated the view that cationic or ionizable lipids are required for efficient RNA uptake, as supported by analysis of delivery methods in clinical trials.[13]

Compared with lipofection, gymnotic uptake is considered a slower and less efficient process.[7] Currently, its study is mostly restricted to the context of short and highly modified therapeutic ASOs. Several factors explain the disadvantage of this uptake route for longer or unmodified RNAs. For example, endosomal escape, the rate-limiting step in RNA therapeutics,[16] is facilitated by an LNP-induced endosomal rupture mechanism that would not apply in the case of gymnosis. Immunological studies also argue against extracellular naked RNAs being functional. For

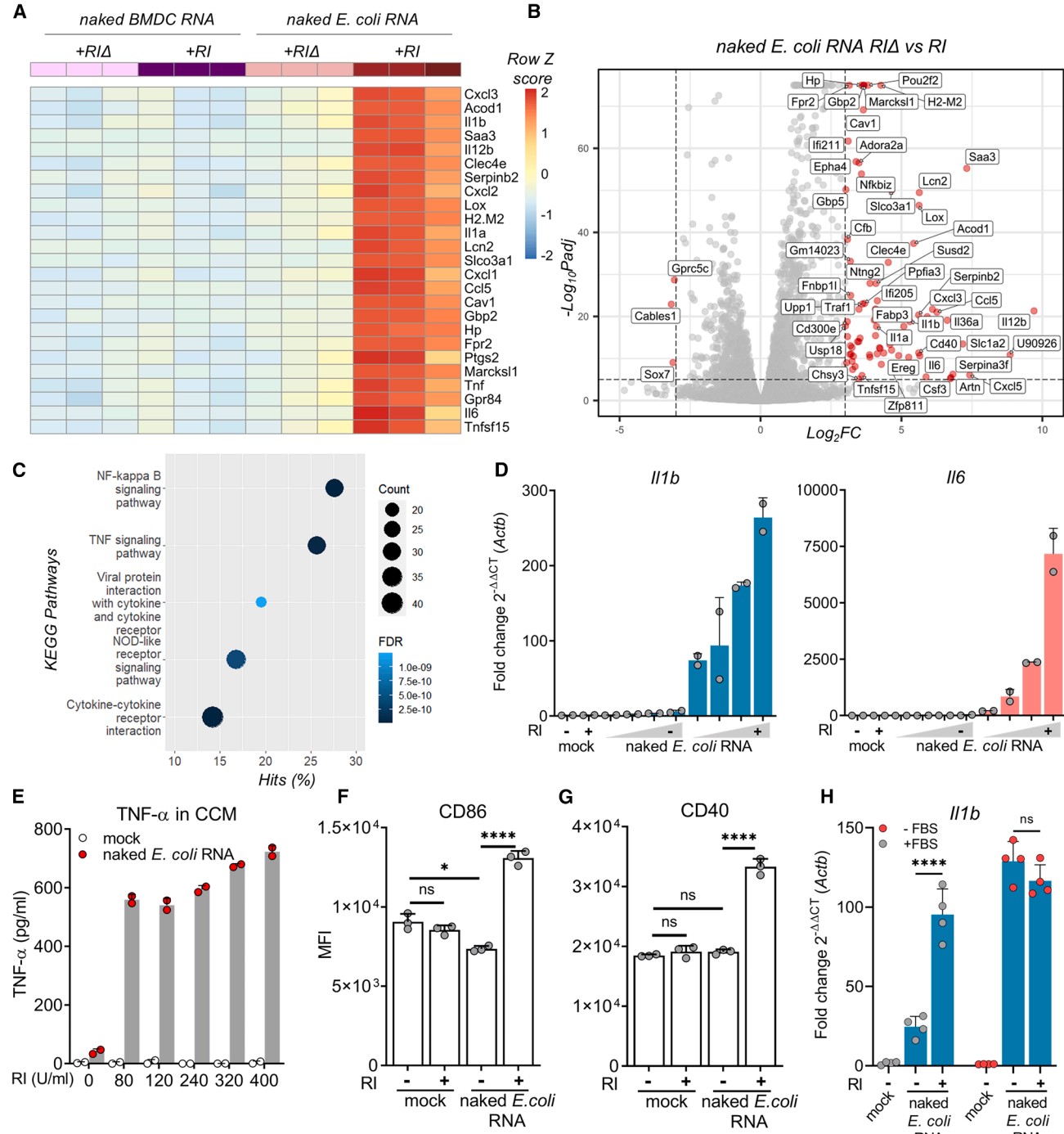

**Figure 1. Naked bacterial exRNA activates BMDCs in an RI-dependent manner**

(A) Heatmap showing top 25 most differentially regulated genes in BMDCs stimulated for 6 h with 100 ng/mL naked RNA from *E. coli* or from BMDCs in the presence of RNase inhibitors (+RI) or a thermally inactivated RI instead (+RIΔ).

(B) Volcano plot of differentially expressed genes (adjusted *p* [p-adj] < 1 × 10⁻³, |log₂FC| ≥ 3) between BMDCs stimulated with naked total *E. coli* RNA (+RI vs. +RIΔ).

(C) Pathway enrichment analysis corresponding to BMDCs stimulated with naked *E. coli* RNA + RI.

(D) *Il-1b* and *il-6* expression measured by RT-qPCR in BMDCs stimulated with varying doses (1, 5, 10, or 25 μg/mL) of naked *E. coli* RNA, with or without RI. DPBS was used as negative control.

(E) TNF-α secreted by BMDCs after stimulation (6 h) with 1 μg/mL naked *E. coli* RNA (or DPBS) with varying doses of RI.

*(legend continued on next page)*

instance, transfection of bacterial RNAs using cationic lipids triggers immune cell activation in human and murine macrophages,[17,18] but this response is abrogated when the RNA is spiked into the media in the absence of lipofection reagents.[17]

Finally, naked RNA is highly unstable in biofluids and extracellular samples, which tend to contain high quantities of active ribonucleases (RNases). This fact, coupled with the low efficiency of gymnotic uptake,[7] and the apparent lack of immune cell activation induced by naked extracellular bacterial RNAs,[17] explains why RNA-mediated intercellular communication is mostly—and almost exclusively—studied in the context of EVs.[19]

We have recently shown that some RNAs are intrinsically stable in the extracellular space and act as stable reservoirs of shorter RNA fragments that can enter cells spontaneously.[20] In addition, when adding a broad-range ribonuclease inhibitor (RI) to human cancer cell-conditioned medium, we stabilized a previously unknown population of extracellular ribosomes that induced dendritic cell maturation.[21] These results suggest that extracellular, nonvesicular RNAs, comprising both naked RNAs and ribonucleoprotein complexes, can interact with recipient cells and induce downstream signaling responses. Importantly, to observe any functional interaction involving nonvesicular extracellular RNAs (exRNAs), strict control of extracellular RNase activity is required. This is impossible in typical cell culture experiments, in which 10% fetal bovine serum (FBS) is usually employed. We hypothesize that failure to control for serum-derived RNase activity has so far precluded the study of intercellular communication pathways mediated by nonvesicular RNAs, hence underestimating the efficiency of gymnosis as a functional and efficient RNA uptake process.

Here, using a broad-spectrum RI, we show that both single-stranded and double-stranded long and short RNAs can be spontaneously internalized by different murine and human cell types in the absence of any transfection reagents. Internalized RNAs can be sensed by Toll-like receptors (TLRs) inside endosomal compartments. Furthermore, at least some RNA molecules can escape the endosomes into the cytosol and regulate or modify gene expression in recipient cells. Intravenous administration of RI enhances the inflammatory responses triggered by naked exRNA *in vivo*. In contrast, compartments with low RNase activity, such as the peritoneal cavity, seem to be inherently prone to nonvesicular exRNA recognition. Altogether, these results suggest that naked or nonvesicular exRNAs and extracellular RNases might work as antagonistic inducers and regulators of compartment-specific inflammatory responses.

## RESULTS

### Naked bacterial exRNA activates BMDCs in an RI-dependent manner

Earlier studies showed lack of exRNA recognition by innate immune cells when bacterial RNA, a well-known pro-inflammatory molecule, was spiked into the medium instead of transfected.[17,22] This suggests that the spontaneous uptake of naked exRNA is inefficient, possibly due to the fact that cells are refractory to internalizing negatively charged nucleic acids.[8] Alternatively, it could simply be a consequence of RNA stability. Because cell-culture experiments are typically done in the presence of FBS, a rich source of RNases, we hypothesized that this could explain the apparent lack of naked-exRNA bioactivity seen in earlier studies.[19–21] To study this, we worked with primary cultures of murine bone-marrow-derived cells (BMDCs), differentiated into antigen-presenting cells.[23,24] We incubated these BMDCs with purified *Escherichia coli* RNA in the absence of any transfection reagents, but adding either a broad-range extracellular RNase inhibitor (+RI) or a thermally inactivated inhibitor (+RIΔ), lacking any protective activity (Figure S1A). Consistent with our hypothesis, naked bacterial exRNA induced a highly pro-inflammatory transcriptional signature only in the presence of RI (Figures 1A–1C, S1B, and S1C). Of note, BMDC-derived self-RNA was immunologically silent regardless of RI addition (Figures 1A and S2).

Previous results were further confirmed by RT-qPCR. All tested genes (*Cxcl10*, *Ifit2*, *Il1a*, *Il1b*, *Il6*, *Oas3*) showed a dose-dependent induction in the presence of RI (Figures 1D and S3). Additionally, pretreatment of naked *E. coli* RNA with RNAse1 completely abrogated the observed effects regardless of RI addition (Figures S1B and S1C). Interestingly, BMDCs exposed to naked *E. coli* RNA released high levels of tumor necrosis factor (TNF)-α (Figure 1E) and increased the expression of cell surface markers associated with DC activation, such as CD40 and CD86 (Figures 1F and 1G), only in the presence of RI.

Interestingly, when previous experiments were repeated in serum-deprived medium (−FBS), the response elicited by BMDCs to naked *E. coli* RNA was as strong as previously observed but now completely independent of RI (Figure 1H). Overall, these results demonstrate that naked bacterial RNA can be specifically sensed by murine BMDCs, but this recognition cannot be observed if extracellular RNases are not previously inhibited or removed.

### RI enhances the endocytic uptake of naked RNA by exRNA stabilization

To understand whether naked-exRNA sensing requires prior internalization, we incubated BMDCs with Dynasore, a dynamin inhibitor that blocks clathrin-dependent endocytosis and macropinocytosis,[25] before the addition of naked exRNAs. Upregulation of *Il6*, *Il1b*, *Cxcl10*, and *Ifit2* in response to bacterial exRNA was strongly inhibited in the presence of the drug (Figure 2A). Of note, although *Il6*, *Il1b*, *Cxcl10*, and *Ifit2* induction was strongly dependent on RI (as previously observed; Figures 1D and S3), endocytosis of Dextran-AF647 was independent of RI (Figure 2B). These results demonstrate that the main mechanism by which RI facilitates naked-exRNA uptake is by stabilizing RNA

---

(F and G) Flow cytometry analysis of CD40 and CD86 in BMDCs stimulated for 24 h with 50 μg/mL naked *E. coli* RNA with or without RI. Median fluorescence intensity of CD86 (F) and CD40 (G) are shown.

(H) *Il1b* expression (RT-qPCR) in BMDCs stimulated with 1 μg/mL naked *E. coli* RNA with or without RI, in either serum-deprived medium (−FBS) or serum-containing medium (+FBS) for 6 h. RPMI was used as mock.

(F–H) ns, not significant; *$p < 0.05$, ****$p < 0.0001$; one-way ANOVA with Tukey's multiple comparison test.

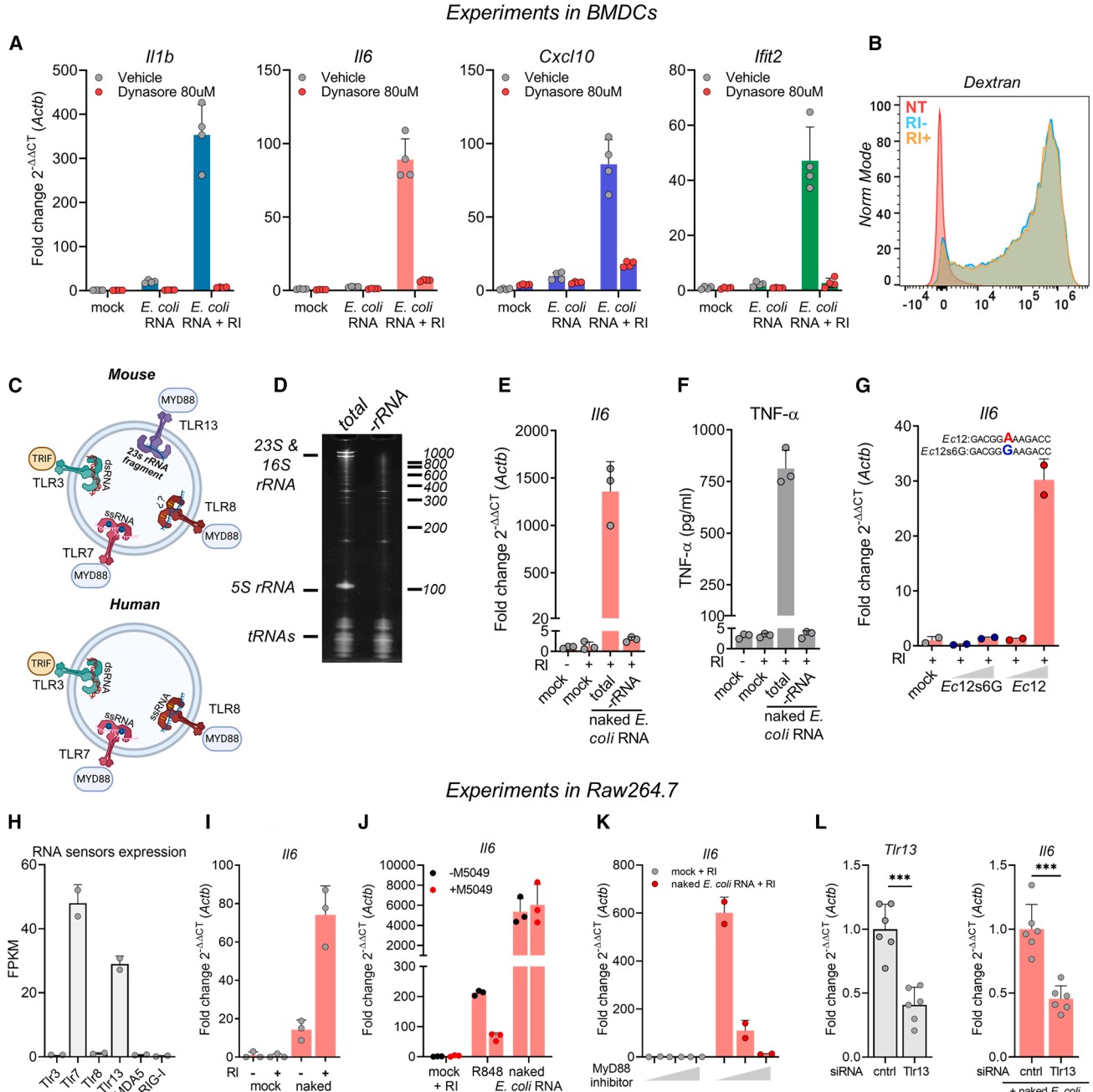

**Figure 2. TLR13 senses naked bacterial ribosomal RNA in BMDCs and Raw264.7 macrophages**

(A) *il6*, *Il1b*, *Cxcl10*, and *Ifit2* expression by RT-qPCR in BMDCs stimulated for 6 h with 1 μg/mL of naked *E. coli* RNA, with or without RI, in the presence of Dynasore (red circles) or vehicle (gray circles).

(B) Flow cytometry analysis of BMDCs untreated or incubated with Dextran-AF647 for 1 h, with or without RI.

(C) Diagram of RNA-sensing endosomal TLRs in mouse (top) and human (bottom). TLR3 senses double-stranded RNA (dsRNA) larger than 40 bp, TLR7 and TLR8 (in humans) sense short single-stranded RNA (ssRNA) in the presence of guanosine (G) or uridine (U), respectively, and TLR13 recognizes an ssRNA fragment derived from 23S bacterial rRNA.

(D) Denaturing PAGE of total and ribosomal RNA-depleted (−rRNA) RNA from *E. coli*.

(E and F) *Il-6* expression (RT-qPCR) and levels of secreted TNF-α (ELISA) of BMDCs stimulated with 1 μg/mL of naked total or rRNA-depleted *E. coli* RNA, in the presence of RI.

(G) *Il6* expression (RT-qPCR) in BMDCs stimulated with 1 or 5 μg/mL of synthetic unmodified naked Ec12 RNA (a known agonist of TLR13), or its mutated version (Ec12s6G), both in the presence of RI. Mock: RPMI.

in the extracellular space, and not by increasing endocytosis rates.

### Naked bacterial RNA is sensed by TLR13 in BMDCs and murine macrophages

The requirement of endocytosis (Figure 2A), and the potential role of the nuclear factor (NF)-κB pathway in BMDC response to naked exRNA (Figure 1C), suggests the involvement of endosomal pattern recognition receptors (PRRs) in these processes (Figure 2C).

To infer which receptor-ligand interaction is mostly responsible for the observed effects, we separated total *E. coli* RNA into a small (<200 nt) and a large (>200 nt) fraction (Figure S4A) and observed that only naked RNAs longer than 200 nt triggered the production of pro-inflammatory cytokines at both the RNA and the protein levels (Figures S4B and S4C). Considering that RNAs longer than 200 nt are mainly ribosomal RNAs (rRNAs), we speculated that extracellular bacterial rRNAs were triggering these effects. Indeed, probe-based selective depletion of bacterial rRNAs (Figure 2D) abrogated both *Il6* transcription and TNF-α release, despite using equal RNA concentrations across conditions (Figures 2E and 2F). This suggests the involvement of TLR13, a mouse-specific endosomal PRR that recognizes bacterial 23S ribosomal RNA and that is expressed in dendritic cells.[22,26]

A 23S rRNA segment, encompassing 12 nucleotides, was previously identified as the ligand of TLR13 in *Staphylococcus aureus*.[22] We first corroborated that this sequence, named here as *Ec12*, is conserved in *Escherichia coli*. Interestingly, BMDCs stimulated with a synthetic and unmodified naked RNA comprising the sequence of *Ec12* triggered *Il6* expression (Figure 2G). Strikingly, this response was completely abrogated when adenosine in position six of *Ec12* was substituted for guanosine (*Ec12s6G*), a substitution that is known to affect TLR13 recognition.[22]

To gain further insights into the potential role of TLR13 in the recognition of naked bacterial RNA, we leveraged the fact that the murine macrophage cell line Raw264.7 expresses TLR7 and TLR13 as the only RNA sensors (Figure 2H). Like BMDCs, Raw264.7 cells can recognize and respond to naked *E. coli* RNA in an RI-dependent manner (Figure 2I and S4D–S4F). Of note, in agreement with previous observations,[27] the human monocytic cell line THP-1, which lacks TLR13, did not respond (Figures S4G–S4I). This lack of response was not due to lack of uptake, as we found that 100% of cells were positive for labeled Ec12 RNA, even when testing concentrations as low as 1 nM (Figure S5).

Interestingly, the response to naked *E. coli* RNA remained unchanged when macrophages were incubated with M5049, a

TLR7/8 specific inhibitor. As expected, M5049 significantly reduced the response to R848, a TLR7/8 agonist (Figure 2J). On the other hand, inhibiting MyD88 in these cells completely abolished the observed response (Figure 2K). Taken together, these results strongly suggest the involvement of the TLR13/MyD88 axis in the response of murine macrophages to naked bacterial RNA. Furthermore, siRNA-mediated knockdown of TLR13 (50% reduction) reduced the response by an almost identical amount (Figure 2L).

Overall, using BMDCs and Raw264.7 macrophages as models, we have shown that naked exRNAs can gain access to the endocytic pathway and trigger activation of endosomal RNA sensors when extracellular RNases are inhibited.

### Naked double-stranded RNA is also recognized by BMDCs

To study whether other nonvesicular exRNAs could also activate BMDCs, we used poly(I:C) as a model to understand naked double-stranded RNA (dsRNA) recognition. This molecule is known to be recognized by TLR3 at the endosomal level and by the RIG-I-like receptors (RLRs) RIG-I and MDA5 in the cytosol when dsRNAs are transfected[28,29] (Figure 3A). Interestingly, when BMDCs were incubated with naked extracellular poly(I:C), there was a strong, RI-dependent increment in the expression of *Cxcl10* and *Ifit2* (Figures 3B and 3C). These two genes were tested since they are known to be induced by poly(I:C) in human DCs (Figures S6A and S6B), and we also validated their induction with naked poly(I:C) in the murine dendritic cell line, JawsII (Figure S6C). In contrast, Raw264.7 macrophages did not respond to naked poly(I:C) (Figures S4D–S4F), in agreement with these cells not expressing any of the above-mentioned RNA sensors (Figure 2H).

### Naked double-stranded RNA is recognized by mitochondrial antiviral-signaling protein-dependent cytosolic RNA sensors in human macrophages

We could not easily determine whether poly(I:C) was being recognized at the endosomal or cytosolic level in BMDCs because these cells express all of the sensors involved in dsRNA recognition (Figure 3D). Human THP-1 monocytes and macrophages could serve as a better model to understand the molecular basis of naked dsRNA recognition because they lack TLR3 but express high levels of RIG-I and MDA5 (Figure 3E). Hence, any response to poly(I:C) in these cells can be attributed, at least in theory, to either of these RLRs.

First, using flow cytometry, we observed that RI enhanced naked fluorescein-labeled poly(I:C) uptake in THP-1 monocytes without altering endocytosis rates, as measured by

---

(H) Expression of cytosolic and endosomal RNA-specific PRRs in Raw264.7 cells. Data were obtained from GEO: GSE103958.

(I) *il6* expression (RT-qPCR) in Raw264.7 stimulated for 6 h with 2 μg/mL of naked *E. coli* RNA, with or without RI.

(J) *il6* expression (RT-qPCR) in Raw264.7 preincubated with or without the TLR7/8 inhibitor M5049 at 1 μM for 3 h and later stimulated for 6 h with 1 μg/mL of naked *E. coli* RNA, with or without RI. The TLR7/8 agonist, R848 at 1 μg/mL, was used as a positive control.

(K) *il6* expression (RT-qPCR) in Raw264.7 preincubated with a Myd88 inhibitory peptide (at 0, 1, or 10 μM) for 4 h, and later stimulated for 6 h with 1 μg/mL of naked *E. coli* RNA, with or without RI.

(L) siRNA-mediated *Tlr13* knockdown in Raw264.7. Left: *Tlr13* expression (RT-qPCR) 48 h post transfection with either a control siRNA at 50 nM or a mix containing two siRNAs directed against *Tlr13* (25 nM each). Right: 48 h post siRNA transfection, Raw264.7 cells were stimulated for 6 h with 1 μg/mL of naked *E. coli* RNA with RI and *il6* expression was measured by RT-qPCR. ***$p < 0.001$; unpaired two-tailed t test.

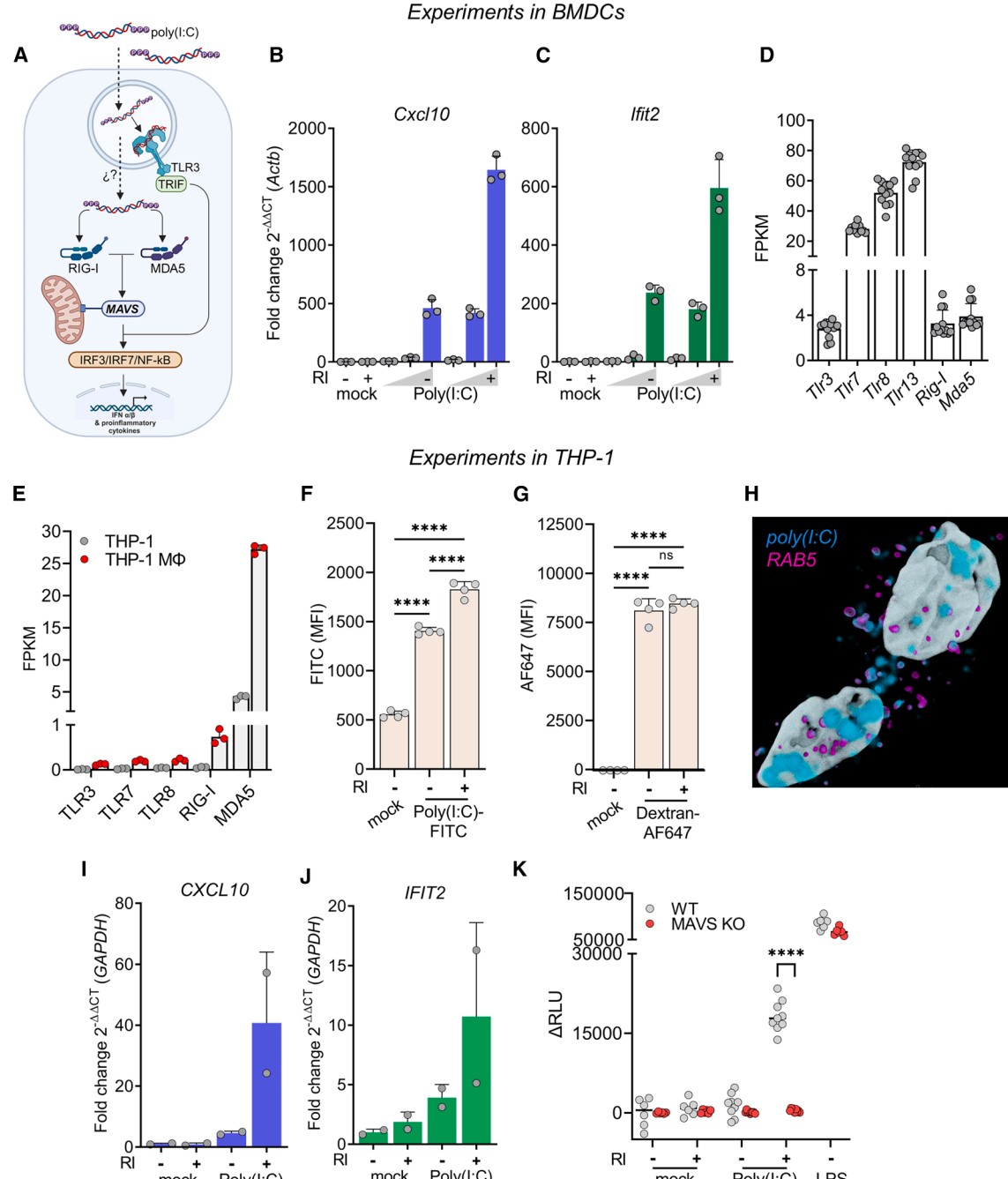

**Figure 3. Naked exRNA triggers MAVS-dependent cytosolic RNA sensors in human macrophages**

(A) Diagram of RNA-sensing endosomal and cytosolic receptor capable of recognizing naked poly(I:C) in BMDCs and THP-1 cells.

(B and C) BMDC expression of *Cxcl10* (B) and *Ifit2* (C) by RT-qPCR after 6-h stimulation with varying doses (0.1, 1, 10 μg/mL) of naked poly(I:C), with (+) or without (−) RI. RPMI was used as negative control.

(D) Expression (fragments per kilobase per million [FPKM]) of cytosolic and endosomal RNA PRRs in BMDCs.

(E) Expression of cytosolic and endosomal RNA-specific PRRs in THP-1 cells or in THP-1-derived macrophages (GEO: GSE130011).

(F and G) Flow cytometry of THP-1 monocytes stimulated for 3 h with naked fluorescein-labeled poly(I:C) at 0.5 μg/mL (F) or dextran conjugated to AF647 (G), both with and without 160 U/mL RI. ns: not significant; ****$p < 0.0001$; one-way ANOVA with Tukey's multiple comparison test.

(H) Confocal microscopy (z stack) and 3D rendering of THP-1 macrophages stimulated for 30 min with naked fluorescein-labeled poly(I:C) at 0.5 μg/mL. Nuclei (white), dsRNA (cyan), and Rab5 (purple) are represented.

*(legend continued on next page)*

AF647-Dextran (Figures 3F, 3G, S7A, and S7B). These cells showed, by confocal microscopy, a labeling pattern consistent with endosomal localization of the labeled RNA (Figure S7C). A similar pattern was also observed in BMDCs (Figure S7D). Importantly, the endosomal localization of internalized poly(I:C) was confirmed in THP-1 macrophages co-stained for Rab5, a well-established early endosome marker (Figures 3H and S7E). Last, in these cells, we also observed a strong *CXCL10* and *IFIT2* induction upon stimulation with naked poly(I:C), but only in the presence of RI (Figures 3I and 3J).

Considering that THP-1 cells lack endosomal TLR3 (Figure 3E), the above results make a strong case for naked poly(I:C) being internalized by endocytosis and then escaping from endosomal compartments into the cytosol to engage cytosolic RLRs. To test this, we used a reporter THP-1 cell line that expresses a secreted luciferase under the control of the interferon regulatory factor (IRF) transcription factors. The IRF pathway is activated in response to type I interferons or several PRR agonists, including RIG-I and MDA5, which are both dependent on the mitochondrial antiviral-signaling protein (MAVS). In agreement with previous results (Figures 3I and 3J), only naked extracellular poly(I:C) in the presence of RI triggered the IRF pathway (Figure 3K). Furthermore, this response was completely abrogated in *MAVS* knockout (KO) cells. This behavior cannot be attributed to a non-responsive KO cell line, as they behaved similarly to wild-type cells when using lypopolysaccharide (LPS), a MAVS-independent IRF pathway inductor.

Overall, these results demonstrate that naked exRNAs can also reach the cytosol, where the MAVS pathway is located, presumably by endosomal escape.

### RI facilitates protein translation after gymnotic uptake of mRNA

Our previous observations suggest that naked exRNA can, in the presence of RI, be internalized by cells and reach the cytosol after escaping the endosomal network. To assess the generalizability of this phenomenon, we studied gymnotic uptake of mRNAs in different human and murine cell types. Because translation occurs exclusively in the cytosol, observation of protein synthesis would provide conclusive evidence of gymnotic uptake, possibly involving endosomal RNA escape. We synthesized capped and polyadenylated *nanoLuc* mRNA by *in vitro* transcription (IVT) and confirmed its translatability using a rabbit reticulocyte extract (Figure 4A). Then, purified *nanoLuc* mRNA was added to murine BMDCs either with or without RI. Surprisingly, when extracellular RNases were inhibited, the intracellular levels of intact full-length mRNA molecules increased by 40-fold (Figure 4B). After 24 h, we detected nanoLuc protein exclusively in BMDCs when extracellular RNases were inhibited (Figure 4C). To further underscore the influence of extracellular RNases on *nanoLuc* mRNA uptake and translation, we subjected BMDCs

to varying concentrations of FBS and RI prior to their incubation with naked extracellular mRNA. As expected, increasing doses of RI stimulated translation, while increasing percentages of FBS had the opposite effect (Figure 4D). Beyond 10% FBS, RNase activity was sufficiently high to abolish *nanoLuc* translation, even in the presence of RI (Figure 4D).

Since BMDCs are professional phagocytic cells, we sought to determine whether gymnosis also occurred in other cell types. Interestingly, several human epithelial cell lines could uptake and translate nanoLuc mRNA in the presence of RI (Figures 4E and 4F). As previously observed in murine BMDCs, nanoLuc translation in human epithelial cells was abrogated in the presence of endocytosis inhibitors (Figure 4G). Additionally, using confocal microscopy, we could detect that U-2 OS cells are also able to capture and translate naked eGFP mRNA, but only if cells are cultured in the absence of FBS and RI is added to the medium (Figures 4H and 4I).

### Compartment-specific RNase activity modulates naked-exRNA-induced inflammation

We have observed that naked exRNA induces pro-inflammatory responses in cultures of human and murine immune cells and that these responses are modulated by extracellular RNase activity. To assess whether naked exRNAs could also induce inflammatory responses *in vivo*, we intravenously injected both naked *E. coli* RNA and poly(I:C) (Figure 5A) with or without 480 units of RI, a dose sufficient to inhibit mouse serum RNases (Figure S8A). Naked exRNAs increased the percentage of activated (CD86+MHCII+) pDCs and macrophages in the spleen (Figure S9), and, more surprisingly, the percentage of activated splenic B cells (CD86+CD69+) and T cells (CD69+) (Figures 5B and S10), with minor or no changes in total cell numbers. Importantly, despite the lower RNase activity observed in mouse serum compared with FBS (Figure S8B), we could still recapitulate the effect of RI addition, which was stronger in the case of poly(I:C). These results are in good agreement with previous findings using LNP-formulated RNAs.[30] We cannot currently determine whether B and T cell activation was cell autonomous due to exRNA recognition by their own innate sensors[31] or requires prior activation of myeloid cells (Figure S9). Nevertheless, these results highlight that naked exRNAs can trigger systemic inflammation *in vivo*.

Our *in vitro* experiments in BMDCs showed that RI is not required for naked-exRNA-induced inflammation when cells are cultured in serum-deprived medium (Figure 1H). This made us wonder whether naked exRNA alone could be sufficient to induce inflammatory responses when released into tissues or compartments with low RNase activity. Using a highly sensitive fluorometric assay, we found that the RNase activity of mouse peritoneal fluid is approximately 4.5-fold lower than an equivalent dilution of mouse serum (Figure 6B). In agreement with this observation, intraperitoneal injection of either naked

---

(I and J) The expression of *CXCL10* (I) and *IFIT2* (J) in THP-1 macrophages, measured by RT-qPCR after 22-h stimulation with naked poly(I:C) at 10 μg/mL, with (+) or without (−) RI. RPMI was used as a negative control.

(K) Relative luciferase activity corresponding to IRF pathway activation in reporter THP-1 dual wild-type or *MAVS* knockout cells after stimulation for 24 h with 5 μg/mL of naked poly(I:C), with or without RI. RPMI and 200 ng/mL LPS were used as negative and positive controls, respectively.

****$p < 0.0001$ Welch ANOVA with multiple comparison Dunnett T3 test.

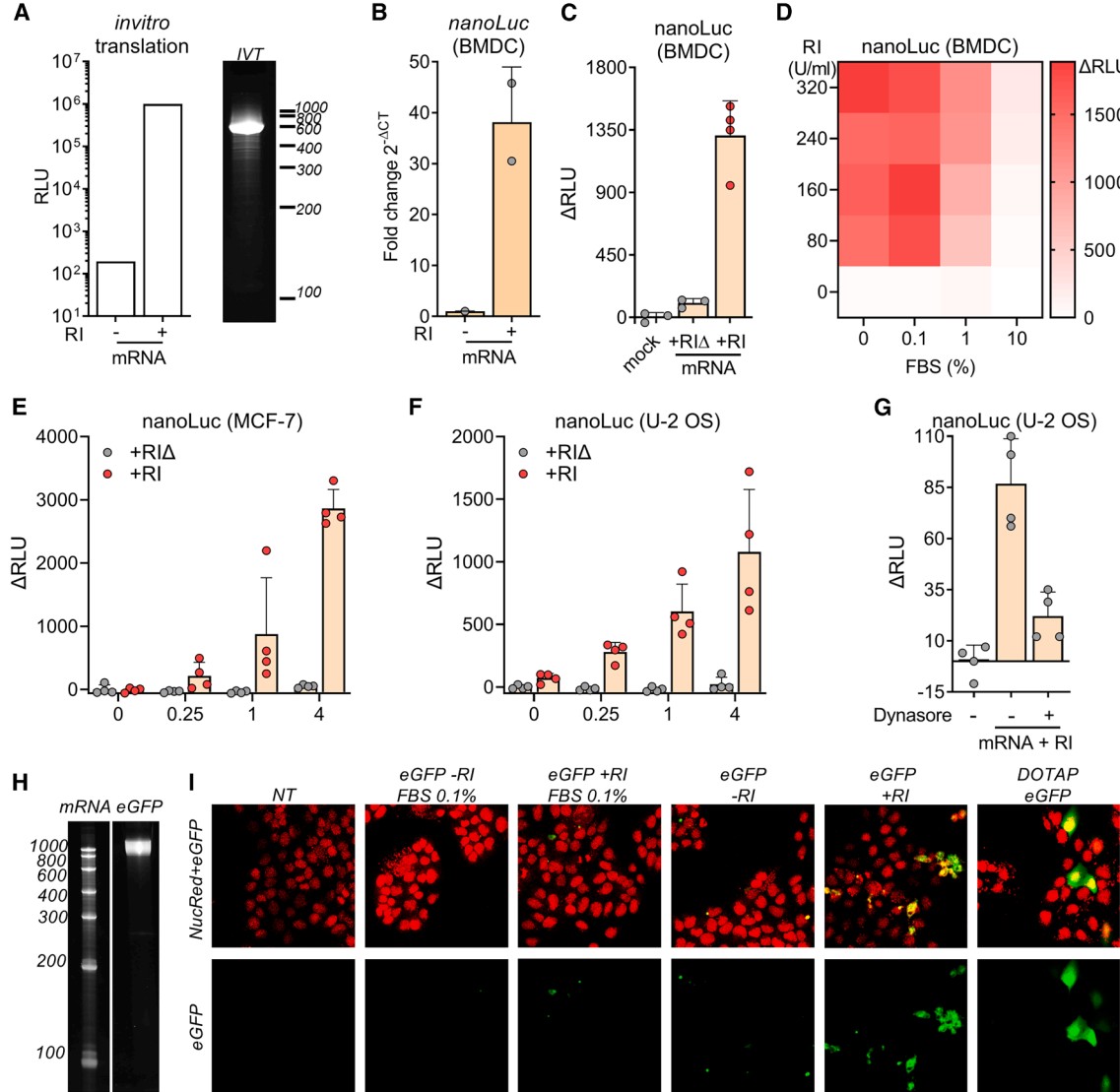

**Figure 4. RI facilitates protein translation after gymnotic uptake of mRNA**

(A) Bioluminescence of *in vitro* translated nanoLuc protein (left) and denaturing PAGE of *in vitro*-transcribed nanoLuc mRNA (right).

(B) BMDCs cultured without FBS were incubated with 1 μg/mL naked nanoLuc mRNA with or without 80 U/mL RI. After 6 h, intact intracellular nanoLuc mRNA was quantified by RT-qPCR.

(C) After 24 h, nanoLuc protein levels were quantified by bioluminescence.

(D) Heatmap showing nanoLuc intracellular protein levels detected by bioluminescence in BMDCs (cultured with varying doses of FBS) as a function of RI concentration.

(E and F) MCF-7 (E) and U-2 OS (F) cells cultured in 0.1% FBS were incubated with increasing doses of nanoLuc mRNA (0, 0.25, 1, 4 μg/mL) with 80 U/mL RI. After 24 h, NanoLuc protein levels were quantified by bioluminescence.

(G) U-2 OS cells cultured in 0.1% FBS were incubated with nanoLuc mRNA and RI in the presence of 80 μM Dynasore or vehicle. After 24 h, nanoLuc protein levels were determined by bioluminescence.

(H) Denaturing PAGE corresponding to *in vitro*-transcribed eGFP mRNA.

(I) Confocal microscopy of U-2 OS cells incubated with 5 μg/mL naked eGFP mRNA with or without 80 U/mL RI at the indicated FBS dose. Non-treated cells and DOTAP-transfected cells were used as negative and positive controls, respectively. Green channel, eGFP; red channel, nuclei.

*E. coli* RNA or poly(I:C) (Figure 6A) triggered a strong inflammatory response that was completely independent of RI (Figures 6C, 6D, and S11). This response was characterized by a strong, dose-dependent disappearance of resident large peritoneal macrophages (LPMs) and recruitment of myeloid cells (comprising monocytes and/or neutrophils) and inflammatory macrophages (iMFs) at 24 h, a hallmark of peritoneal inflammation.[32] Of note, the recruitment of monocytes/neutrophils was not observed in the case of poly(I:C) treatment, which deserves further investigation.

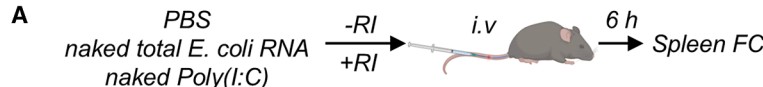

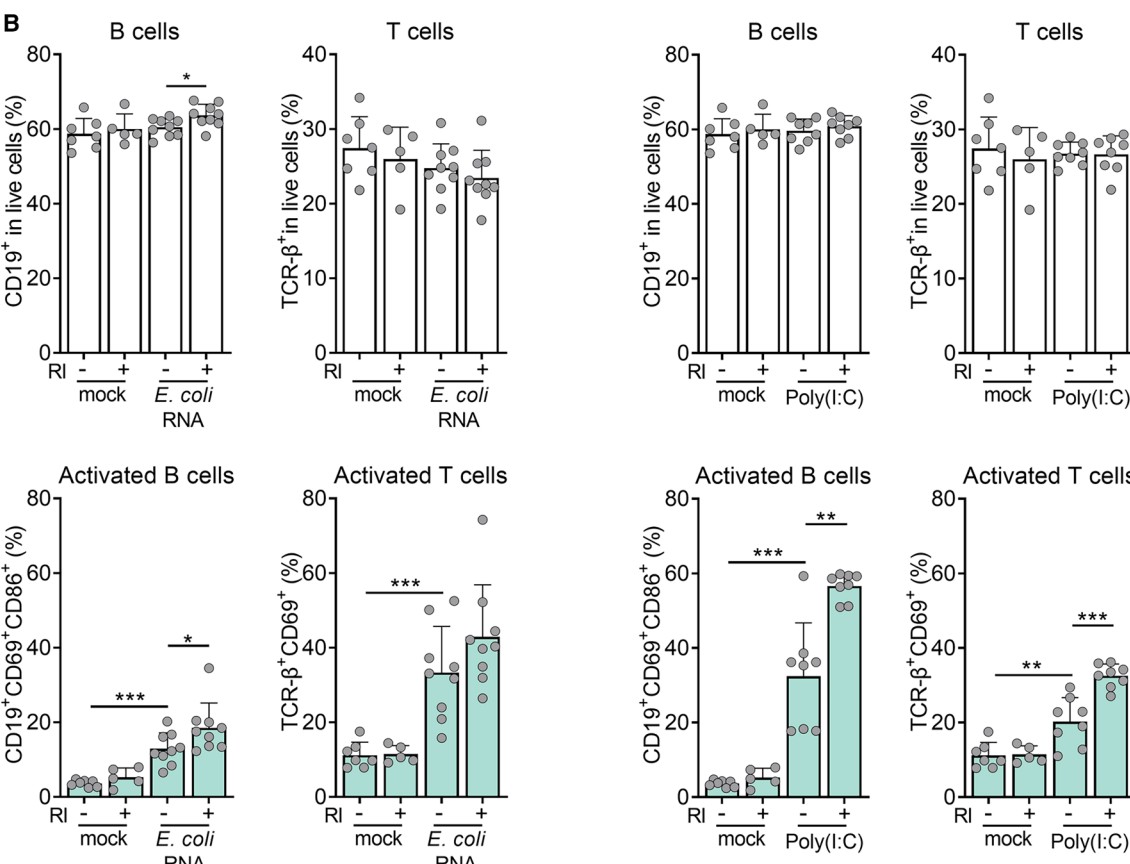

**Figure 5. RI enhances the immunostimulatory effects of naked exRNAs circulating in the blood**

(A) Experimental scheme.

(B) Flow cytometry analysis of splenic total B cells (defined as CD19[+]), total T cells (defined as T cell receptor [TCR]-β[+]), activated B cells (defined as CD19[+]CD69[+]CD86[+]), and activated T cells (defined as TCR-β[+]CD69[+]) 6 h after intravenous administration of naked poly(I:C) (1 μg) or naked total *E. coli* RNA (10 μg), with or without 480 U of RI. Mock: DPBS. Total cells are shown in the top panel, and activated cells are shown in the bottom panel. *$p < 0.05$, **$p < 0.01$, ***$p < 0.001$; unpaired two-tailed t test (if normally distributed data) or unpaired two-tailed Mann-Whitney test (when data had no normal distribution) [naked RNA - RI vs. naked RNA + RI], [naked RNA − RI vs. mock − RI]. See also Figures S9 and S10.

In summary, naked exRNA is inherently bioactive in RNase-poor environments such as the peritoneal cavity, while RI is needed for maximum activity in the bloodstream. Altogether, these results suggest a key role for bloodborne extracellular RNases in the regulation of exRNA-induced systemic inflammation.

## DISCUSSION

RNA-mediated intercellular communication has been almost exclusively studied in the context of EVs,[5] despite most exRNAs being present outside EVs in human biofluids,[33,34] cell-conditioned media,[21,35,36] and even in plants.[37,38] A bias against the study of nonvesicular exRNAs can be explained by the wide-spread assumption that exRNA is unstable unless protected inside EVs. Additionally, EVs provide a mechanism for functional RNA delivery into recipient cells.[5] In contrast, naked RNA is assumed to be incapable of penetrating the "billion-year-old barrier" comprising the plasma membrane and the membrane of endocytic vesicles.[8] By being refractory to the uptake of exRNAs, cells could protect themselves from selfish genetic elements such as viroids and positive-stranded RNA viruses, while preserving their transcriptional identity.

However, several RNA species are stable in the extracellular space even when not associated with EVs. These resilient exRNAs include ribonucleoprotein particles (RNPs) such as Ago2/miRNA complexes,[33,34,39] protein-protected RNA

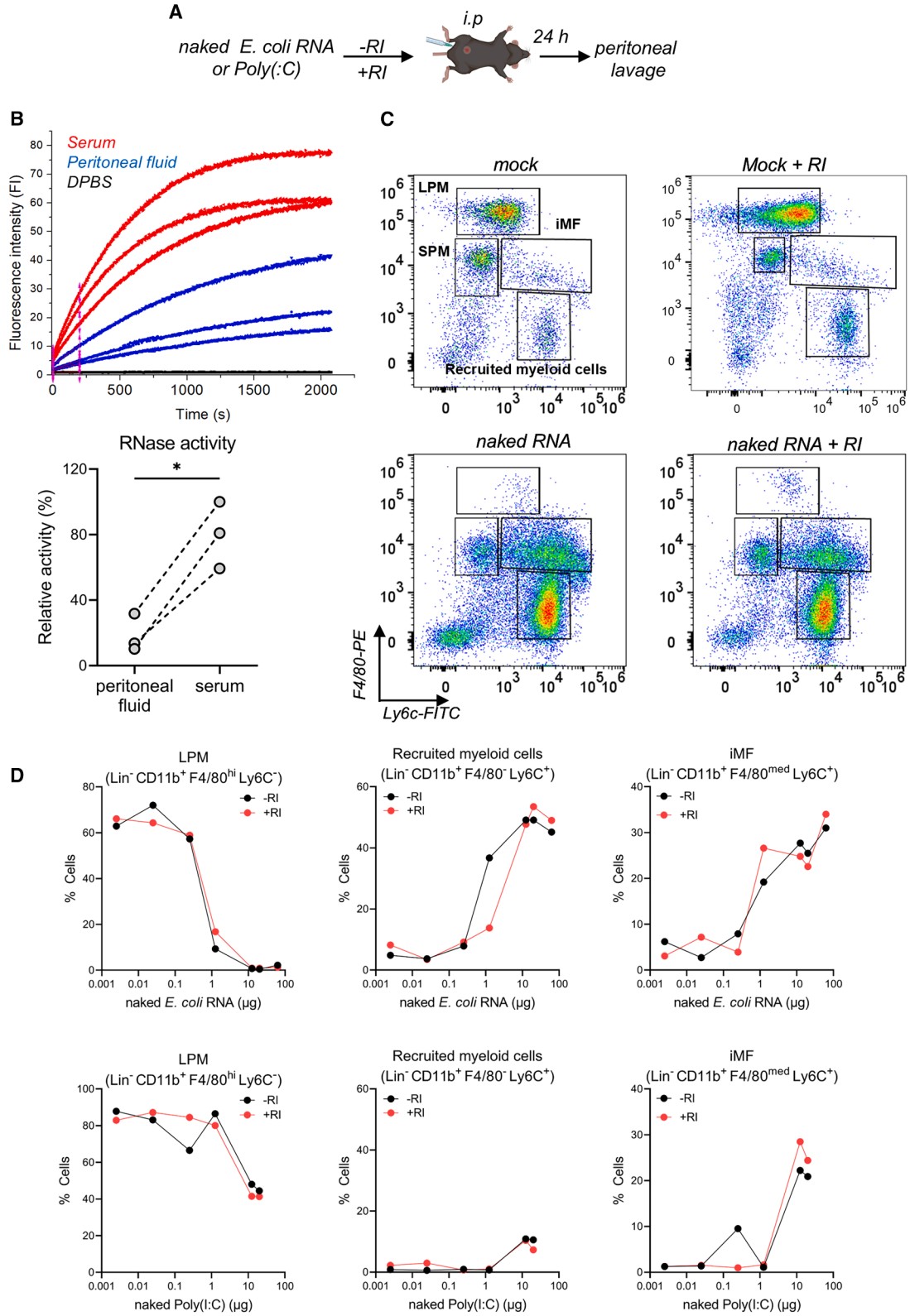

*(legend on next page)*

fragments[40] derived from the ribosome,[20,21,41] U2 snRNPs,[41] and possibly Xist RNPs.[42] Additionally, a population of intrinsically stable nicked tRNAs circulates in human biofluids, probably unprotected or naked.[20]

Rather than being inert, naked exRNAs can enter cells by gymnosis. It should be noted that this term was not defined in the context of a precise molecular mechanism[7]; the name axiomatically refers to functional naked RNA delivery into the cytosol.[43] Since best results in the original publication were obtained with locked nucleic acid (LNA)-containing phosphorothioate ASOs,[7] gymnotic uptake was later considered as an emerging property of phosphorothioate-containing oligonucleotides.[8] Unmodified and longer naked RNAs are thought to lack any biological activity.

In this work, we provide a different interpretation: any naked RNA can enter cells by endocytosis, and even reach the cytosol, as long as its extracellular stability is sufficiently high. The main factor limiting the concentration of naked exRNAs are extracellular RNases, which are highly abundant and active in biofluids such as the human blood.[20,44] Phosphorothioate-containing ASOs are resistant to degradation, and this may be why gymnotic uptake can occur for ASO-based biotherapeutics.[6] By adding a broad-range RNase inhibitor to cell culture medium, we demonstrated that uptake can be extended to naked bacterial rRNAs, synthetic short oligonucleotides (e.g., the Ec12 sequence), poly(I:C), and in vitro-transcribed mRNAs. This was shown in a variety of different cell types, including primary cultures of professional phagocytes and human epithelial cell lines.

It should be noted that we used the term "naked RNA" throughout this study because all stimuli were added in the absence of proteins or transfection reagents, including lipid nanoparticles. However, we cannot rule out the possibility that these naked RNAs might form a complex with one or more soluble extracellular protein(s) or other binding partners before cellular uptake. Having said that, the fact that similar responses were obtained in experiments performed in FBS-containing and FBS-free fresh media (Figure 1H) argues against this possibility.

We have shown that naked exRNAs, when stabilized by RI, are intrinsically bioactive in recipient cells. They can engage with endosomal RNA sensors such as TLR13 (in BMDCs and Raw264.7 macrophages), but they can also escape from endosomes into the cytosol, where they can be recognized by MAVS-dependent cytosolic RNA sensors (at least in human THP-1 macrophages). More strikingly, two different mRNAs encoding reporter proteins could be internalized by gymnosis and still be capable of serving as templates for protein synthesis in both BMDCs and human epithelial cell lines. This suggests their capacity to perform endo-

somal escape because ribosomes are present only in the cytosol. It is important to mention that these mRNA translation assays are highly sensitive to extracellular RNases, because a single cleavage event would render an mRNA untranslatable. Hence, the effect of RI was more pronounced in these assays and was potentiated by FBS depletion.

RNA endosomal escape in the absence of LNPs means that the billion-year-old barrier[8] may actually be leaky under certain circumstances. Interestingly, while efficient translation of our GFP-encoding mRNA required the use of modified nucleotides to avoid recognition by TLR7,[45] nanoLuc mRNA was translated even with unmodified uridine. These results encourage further basic research in naked mRNA therapies and vaccines. Although positive results were obtained after localized naked mRNA injections in organs with presumably low extracellular RNase activity,[9–11] co-administration of RI could enable more systemic delivery routes.

If pathogen-derived naked exRNAs are potent pro-inflammatory molecules, how are potentially life-threatening systemic inflammatory responses avoided? For example, the complement system could induce bacterial lysis in the bloodstream, releasing high loads of pro-inflammatory bacterial rRNAs that could activate circulating leukocytes. This study strongly suggests that extracellular RNases are highly abundant in the bloodstream to avoid RNA-induced systemic inflammation. We showed this by demonstrating that RI co-administration enhances immune cell activation in the spleen after intravenous injection of naked RNA. Interestingly, RI was dispensable when a similar assay was performed in the peritoneal cavity, an important site of immune surveillance with much lower RNase activity. Thus, naked exRNAs might convey important information for immune cells residing in tissues with relatively low blood irrigation and hence with low extracellular RNase content.

The immune system has multiple regulatory layers to ensure an optimal response to the different stimuli that immune cells could encounter. One key challenge is to differentiate self from non-self nucleic acids, which is achieved by regulating both the affinity[46] and the subcellular localization of endosomal TLRs.[47,48] Indeed, we observed a dramatically different response when comparing bacterial and murine RNAs incubated at the same concentration, both in the presence of RI. Surprisingly, the effect of RI itself was as dramatic as the capacity of BMDC RNA sensors (predominantly TLR13) to discriminate self from non-self RNAs. Thus, RNase-mediated control of exRNA stability lies on top of a series of layers comprising control of endocytosis, TLR loading and recognition, endosomal escape, and engagement with cytosolic RNA sensors.

---

**Figure 6. Naked exRNA is intrinsically bioactive in the peritoneal cavity**

(A) Experimental scheme.

(B) Comparison of RNase activity in serum and peritoneal fluid of three mice using a fluorometric assay. Top: kinetic curves. Bottom: the relative RNase activity derived from initial reaction velocities. *$p < 0.05$; paired two-tailed t test.

(C) Representative flow cytometry dot plots showing expression of F4/80 and Ly6c (gated on lineage⁻ CD11b⁺) to characterize immune cell populations in the peritoneal cavity 24 h after intraperitoneal (i.p.) injection of DPBS or 25 μg of naked total *E. coli* RNA, with or without 40 U of RI. Resident large (LPM) and small (SPM) peritoneal macrophages, recruited myeloid cells and inflammatory macrophages (iMFs) are shown. See also Figure S11.

(D) Graphs showing the percentage of resident large peritoneal macrophages (LPMs) defined as Lin⁻CD11b⁺F4/80^hiLy6c⁻, recruited myeloid cells defined as Lin⁻CD11b⁺F4/80⁻Ly6c⁺, and inflammatory macrophages defined as Lin⁻CD11b⁺F4/80^medLy6c⁺, 24 h after i.p. injections of increasing doses of naked total *E. coli* RNA (top panel, 0.025, 0.250, 1.25, 12.5, 25, and 62.5 μg) or naked poly(I:C) (bottom panel, 0.0025, 0.025, 0.250, 1.25, 12.5, and 20 μg) in the presence or absence of 40U of RI. The lowest concentration of naked *E. coli* RNA corresponds to DPBS, adjusted to fit the log scale.

One interesting finding that deserves further discussion is that the uptake of Ec12 RNA, unlike poly(I:C), did not depend on RI addition (Figures S5 and 3F, respectively). Moreover, the uptake efficiency in FBS-containing medium was almost identical when the Ec12 sequence was presented in the context of a single-stranded RNA or a double-stranded DNA, although the latter is presumably more stable (Figure S5). These results imply that the Ec12 RNA is inherently stable against RNase A family members, which might be explained by its pyrimidine-poor sequence composition, lacking any uridine residues. This observation made us wonder about the evolution of RNA sensors. For example, TLR13 is known to recognize a conserved sequence motif in bacterial 23S rRNA,[22] but our results suggest that this specific sequence might have also been selected based on its extracellular stability. This conclusion would be consistent with RNA-sensing TLRs recognizing mainly cell-free exRNAs.

There are several lines of evidence pointing to a role of enhanced and sustained exRNA sensing as a trigger of systemic inflammation or autoimmunity.[31,42,46,47,49–51] It is striking that all single-stranded endosomal RNA sensors (i.e., TLR7, TLR8, and TLR13) are codified in the X chromosome, possibly explaining sex-biased autoimmunity due to defects in dosage compensation.[52] These studies demonstrate that endosomal RNA sensing must be exquisitely fine-tuned[48] and that the control of TLR expression and localization is an effective way of doing so. Our study strongly suggests that extracellular RNases may act as an additional regulatory layer to control the levels of pro-inflammatory exRNAs and that they should therefore be considered as a regulatory arm of the innate immune system.

Supporting the role of extracellular RNases in the regulation of inflammatory responses, several reports have shown that ectopic administration of extracellular RNases reduces inflammation in mouse models of lupus-like disease,[53] traumatic brain injury,[54] and myocardial infarction.[55] In humans, sterile inflammation causing cell death and the release of RNPs into the extracellular space seems to be an important trigger of systemic lupus erythematosus.[42,56] In this and other autoimmune diseases, clinical trials using RNase-based drugs have shown evidence of efficacy.[57,58]

On a more technical note, we would like to emphasize that intents to understand exRNA biology using cell-culture-based assays will inevitably fail if fetal or calf serum is used without RI supplementation. This technical problem has led to the wrong assumption that exRNA biology is relevant only in the context of EVs, or that the study of intracellular nucleic acid sensors requires the use of transfection reagents.[17] We envision that this study will renovate interest in nonvesicular exRNAs in intercellular communication, particularly in the context of inflammatory and autoimmune diseases.

In conclusion, this study clearly demonstrates that naked exRNA is intrinsically bioactive both *in vitro* and *in vivo*. Moreover, in the absence of RNases, exRNA can spontaneously enter cells and even escape from endosomal compartments, engaging both endosomal and cytosolic RNA sensors. Earlier studies may have failed to observe these phenomena due to the confounding effect of extracellular RNases. Furthermore, this study suggests that the multiple RNases that exist in the extracellular space may have evolved to avoid potentially harmful systemic inflammatory responses triggered by either host- or pathogen-derived naked or nonvesicular exRNAs.

### Limitations of the study

Although this study has several strengths, including the use of diverse cellular models, multiple naked RNA species, and the combination of *in vitro* and *in vivo* results, a key limitation is the reliance on exogenous RNAs (lacking RNA modifications) for *in vivo* studies. Future work should complement these findings by modulating endogenous exRNA levels. Additionally, findings on TLR13 activation in murine cell lines may not directly translate to humans, as this RNA sensor is not expressed in primates.

### RESOURCE AVAILABILITY

#### Lead contact
Further information and requests for resources and reagents should be directed to and will be fulfilled by the corresponding author, Juan Pablo Tosar (jptosar@pasteur.edu.uy).

#### Materials availability
This study did not generate new unique reagents.

#### Data and code availability
The sequencing data generated in this study can be accessed at NCBI under BioProject PRJNA1120732 (SRA: SRP512141).

### ACKNOWLEDGMENTS

The authors gratefully acknowledge the following core facilities and technological platforms at the Institut Pasteur of Montevideo for their support and assistance in the present work: Advanced Bioimaging Unit (UBA), Animal Biotechnology Unit (UBAL), and the Cell Biology Unit (UBC), with special thanks to Karen Perelmuter and Paula Céspedes. The authors want to thank María Elena Márquez, Sofía Russo, Daniela Olivera, Germán Galliussi, and Valentina Perez for helpful suggestions, experimental help, and insightful scientific discussions. This study was funded by Agencia Nacional de Investigación e Innovación (ANII, Uruguay) (FCE_1_2021_1_166344; PEC_3_2019_1_158011), Fondo para la Convergencia Estructural del Mercosur (FOCEM) (COF 03/11), Comisión Sectorial de Investigación Científica (CSIC, UdelaR, Uruguay) (22620220100069UD), Comisión Académica de Posgrado (CAP, CSIC, UdelaR, Uruguay) (BDDX_2020_1#50810037; BFPD_2023_1#50810037), the National Institutes of Health (NIH, USA) (R21CA263424), and the NIH Office of the Director (UH3CA241694).

### AUTHOR CONTRIBUTIONS

J.P.T. and M.S. led and oversaw the study. J.P.T., M.S., and M.C. conceived the project and designed experiments. M.C. performed and analyzed most *in vitro* and *in vivo* experiments. M.C. carried out bioinformatic analysis. V.B., B.C., and M.L.C. conducted and analyzed naked mRNA capture experiments, RNA stability experiments, and confocal microscopy experiments, respectively. K.W., M.H., and A.C. contributed to data interpretation and evaluation. J.P.T. and M.C. wrote the manuscript with input from all authors.

### DECLARATION OF INTERESTS

J.P.T., M.S., M.H., and A.C. are members of the Sistema Nacional de Investigadores (SNI) and Programa de Desarrollo de las Ciencias Básicas (PEDECIBA, MEC-UdelaR). K.W. has a sponsored research agreement with Ionis Pharmaceuticals; is or has been an advisory board member of ShiftBio, Exopharm, NeuroDex, NovaDip, and ReNeuron; and performs *ad hoc* consulting as Kenneth Witwer Consulting. J.P.T. is a founder of B4-RNA, a startup involved in exRNA-based diagnosis. J.P.T., B.C., A.C., and K.W. have filed patents on RNA repair and sequencing for biomarker/diagnostic applications.

After submitting the initial version of this manuscript, J.P.T. was contacted and later became a member of the advisory board of Resolve Therapeutics, a company employing synthetic RNases to treat human diseases. This company had no involvement in this study.

## STAR★METHODS

Detailed methods are provided in the online version of this paper and include the following:

- KEY RESOURCES TABLE
- EXPERIMENTAL MODEL DETAILS
  - Animals
  - Cell lines
  - Primary cell culture
- METHOD DETAILS
  - Total E. coli RNA purification and fractionation
  - RNA-seq and bioinformatic analysis
  - Bioinformatic analysis of publicly available datasets
  - Cell stimulation
  - RT-qPCR
  - Tlr13 knockdown in Raw264.7
  - *In vitro* transcription
  - *In vitro* translation
  - RNase activity in mouse serum and peritoneal wash
  - nanoLuc mRNA internalization and translation assays
  - Naked eGFP internalization and translation assays
  - *In vivo* experiments
  - Flow cytometry
  - RNA polyacrylamide electrophoresis
- QUANTIFICATION AND STATISTICAL ANALYSIS

## SUPPLEMENTAL INFORMATION

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

## STAR★METHODS

### KEY RESOURCES TABLE

| REAGENT or RESOURCE | SOURCE | IDENTIFIER |
|---|---|---|
| **Antibodies** | | |
| FITC anti-mouse CD40 | BioLegend | Cat# 124607; RRID:AB_1134090 |
| PE anti-mouse CD86 | BD Biosciences | Cat# 553692; RRID:AB_394994 |
| APC anti-mouse MHCII(I-A/I-E) | BioLegend | Cat# 107614; RRID:AB_313329 |
| APC/Cyanine7 anti-mouse CD19 | BioLegend | Cat# 115529; RRID:AB_830706 |
| APC/Cyanine7 anti-mouse TCR beta chain | BioLegend | Cat# 109220; RRID:AB_893624 |
| Brilliant Violet 711(TM) anti-mouse Ly-6G | BioLegend | Cat# 127643; RRID:AB_2565971 |
| PE-Cy7 anti-mouse CD11c | BD Biosciences | Cat# 558079; RRID:AB_647251 |
| PerCP/Cyanine5.5 anti-mouse/human CD45R/B220 | BioLegend | Cat# 103236; RRID:AB_893354 |
| PE anti-mouse F4/80 | BD Biosciences | Cat# 565410; RRID:AB_2687527 |
| Alexa Fluor(R) 700 anti-mouse/human CD11b | BioLegend | Cat# 101222; RRID:AB_493705 |
| FITC anti-mouse CD86 | BioLegend | Cat# 105006; RRID:AB_313149 |
| PerCP/Cyanine5.5 anti-mouse/human CD11b | BioLegend | Cat# 101228; RRID:AB_893232 |
| PE-Cy7 anti-mouse CD69 | BD Biosciences | Cat# 552879; RRID:AB_394508 |
| Anti-human RAB5A | Cell Signaling | Cat# 46449; RRID:AB_2799303 |
| Alexa Fluor(R) 488 anti-mouse IgG (H + L) | Invitrogen | Cat# A-11029; RRID:AB_2534088 |
| Alexa Fluor(R) 647 anti-mouse IgG (H + L) | Invitrogen | Cat# A-21236; RRID:AB_2535805 |
| **Chemicals, peptides, and recombinant proteins** | | |
| 3′-O-Me-m7G(5′)ppp(5′)G RNA Cap Structure Analog | New England Biolabs | Cat #S1411 |
| Acetone | Dorwill | Cat #UN1090 |
| Albumin fraction V pH = 7.0 | AppliChem | Cat #A1391 |
| Ammonium chloride | Sigma-Aldrich | Cat #A9434 |
| Blasticidine | Invivogen | Cat #ant-bl |
| Chloroform | Droguería Industrial Uruguay | Cat #41191 |
| Collagenase D | Roche | Cat #11088858001 |
| DAPI | Sigma-Aldrich | Cat #D9542 |
| Dextran conjugated to Alexa Fluor 647 | Invitrogen | Cat #D22914 |
| DMEM, high glucose, pyruvate | Gibco | Cat #11995065 |
| DMSO | Sigma-Aldrich | Cat #D2650 |
| DNasa I | Roche | Cat #04716728001 |
| DOTAP | Roche | Cat #11202375001 |
| DPBS without $Ca^{2+}$ $Mg^{2+}$ | Gibco | Cat #14190086 |
| Dithiothreitol | Sigma-Aldrich | Cat #D0632 |
| EDTA disodium salt | Sigma-Aldrich | Cat #E6635 |
| EDTA for molecular biology | New England Biolabs | Cat #7011V |
| Ethanol absolute anhydrous | Carlo Erba | Cat #4146052 |
| FastStart Universal SYBR Green Master | Roche | Cat #FSUSGMMRO |
| Fetal bovine serum, qualified Brazil | Gibco | Cat #12657029 |
| HEPES | Gibco | Cat #15630130 |
| L-Glutamine | Gibco | Cat #A2916801 |
| LIVE/DEAD(TM) Fixable Violet Dead Cell Stain | Invitrogen | Cat #L34964 |
| Lipopolysaccharides from Escherichia coli O111:B4 | Sigma-Aldrich | Cat #L4391 |
| MEGM | Lonza | Cat #CC-4136 |
| MEM Non-Essential Amino Acids Solution | Gibco | Cat #11140035 |
| N1-Methylpseudouridine-5′-Triphosphate | TriLink Biotechnologies | Cat #N-1081 |

*(Continued on next page)*

*Continued*

| REAGENT or RESOURCE | SOURCE | IDENTIFIER |
|---|---|---|
| Normal rat serum | Invitrogen | Cat #10710C |
| Normicin | Invivogen | Cat #ant-nr |
| NucRed Live 647 ReadyProbes Reagent | Invitrogen | Cat #R37106 |
| Optimem | Gibco | Cat #51985091 |
| Penicillin-Streptomycin | Gibco | Cat #15140148 |
| Phorbol 12-Myristate 13-Acetate | Sigma-Aldrich | Cat #P8139 |
| Phusion Polymerase | Thermo Fischer | Cat #F-530XL |
| Polymyxin B | Sigma-Aldrich | Cat #P4932 |
| Potassium bicarbonate | Sigma-Aldrich | Cat #60339 |
| QuantiLuc | Invivogen | Cat #rep-qlc |
| R848 | Invivogen | Cat #tlrl-r848 |
| Recombinant Human Ribonuclease, Pancreatic/RNASE1 | Bon Opus Biosciences | Cat #CA30-10UG |
| RLT buffer | Qiagen | Cat #79216 |
| RNA loading dye | New England Biolabs | Cat #B0363 |
| RNase Inhibitor, Murine | New England Biolabs | Cat #M0314 |
| RPMI 1640 glutaMax | New England Biolabs | Cat #M0314 |
| Sodium acetate | Sigma-Aldrich | Cat #S2889 |
| Sodium pyruvate | Gibco | Cat #11360070 |
| SYBR. Gold Nucleic Acid Gel Stain | Invitrogen | Cat #S11494 |
| Trizol | Invitrogen | Cat #15596026 |
| Trypsin-EDTA 0.05% | Gibco | Cat #25300062 |
| Turbo DNase I | Invitrogen | Cat #AM2238 |
| UltraPure TBE Buffer, 10X | Invitrogen | Cat #15581044 |
| Zeocine | Invivogen | Cat #ant-zn |
| $\beta$-mercaptoethanol | Sigma-Aldrich | Cat #M3148 |
| Critical commercial assays | | |
| NEBNext rRNA Depletion Kit | New England Biolabs | Cat #E7850 |
| Bio-Glo-NL Luciferase Assay System | Promega | Cat #J3081 |
| Compensation Beads | Biolegend | Cat #424602 |
| HiScribe T7 High Yield RNA Synthesis Kit | New England Biolabs | Cat #E2040 |
| M-MLV reverse transcriptase | Thermo Fischer | Cat #28025013 |
| Monarch RNA Cleanup Kit | New England Biolabs | Cat #T2040 |
| NEBNext rRNA Depletion Kit | New England Biolabs | Cat #E7850 |
| PureLink quick gel extraction kit | Invitrogen | Cat #K210012 |
| Retic Lysate IVT Kit | Thermo Fischer | Cat #AM1200 |
| TNF-α mouse ELISA Kit | Biolegend | Cat #430904 |
| RNaseAlert Lab Test Kit | Thermo Fischer | Cat #AM1964 |
| Deposited data | | |
| Raw and processed bulk RNA-seq data | This paper | SRA: SRP512141 |
| Raw and processed bulk RNA-seq data | Zhang et al.[68] | GEO: GSE130011 |
| Raw and processed bulk RNA-seq data | Hartveit and Thunold[69] | GEO: GSE125817 |
| Raw and processed bulk RNA-seq data | Galaxy Community[70] | GEO: GSE103958 |
| Experimental models: Cell lines | | |
| MCF-7 | ATCC | Cat#HTB-22 |
| U2-OS | ATCC | Cat#HTB-96 |
| THP-1 | ATCC | Cat#TIB-202 |
| JAWSII | ATCC | Cat#CRL-3612 |
| RAW 264.7 | ATCC | Cat#TIB-71 |
| THP-1 Dual wild type | Invivogen | Cat#thpd-nfis |

*Continued*

| REAGENT or RESOURCE | SOURCE | IDENTIFIER |
|---|---|---|
| THP-1 Dual Mavs knockout | Invivogen | Cat#thpd-komavs |
| **Experimental models: Organisms/strains** | | |
| E. coli 5-alpha Competent cells | New England Biolabs | Cat #C2987H |
| **Oligonucleotides** | | |
| RNA sequence Ec12: 5′-GrArCrGrGrArArArGrArCrC-3′ | This paper | N/A |
| RNA sequence Ec12s6G: 5′-GrArCrGrGrGrArArGrArCrC-3′ | This paper | N/A |
| RNA sequence Ec12 conjugated toTye665: 5'-/Tye665/-GrArCrGrGrArArArGrArCrC-3′ | This paper | N/A |
| Ec12 ssDNA forward sequence conjugated to Cy5: 5'-/Cy5/-GACGGAAAGACC-3′ | This paper | N/A |
| Ec12 ssDNA reverse sequence: 5′-GGTCTTTCCGTC-3′ | This paper | N/A |
| DsiRNA negative control siRNA | IDT | Cat# 51-01-14-04 |
| DsiRNA 1 against TLR13: 5′-GrCrCrCrCrArArCrUrUrArArArArGrCrUrU-3′ 5′-CrArGrArUrUrCrArArGrCrUrUrUrUrArArG-3′ | IDT | Cat#mm.Ri.Tlr13.13.1 |
| DsiRNA 2 against TLR13: 5′-CrUrUrGrArArGrGrUrCrArUrUrArArArUrCrA-3′ 5′-ArUrGrArCrUrUrUrGrArUrUrArArUrGrArC-3′ | IDT | Cat#mm.Ri.Tlr13.13.2 |
| Poly(I:C) (HMW) Fluorescein | Invivogen | Cat#tlrl-picf |
| Poly(I:C) HMW | Invivogen | Cat#tlrl-pic |
| DNA primer sequences for qPCR | This paper | See Table S1 |
| DNA primer sequences for IVT template amplification | This paper | See Table S2 |
| **Recombinant DNA** | | |
| pEGFP-N2 plasmid | Clontech TaKaRa | N/A |
| pUAS-NanoLuc plasmid | Zhang et al.[68] | Addgene Cat #87696 |
| **Software and algorithms** | | |
| FlowJo v10.8.1 | LLC | https://www.flowjo.com |
| SnapGene | N/A | https://www.snapgene.com/ |
| Prism 9 | Graphpad | https://www.graphpad.com |
| ImageJ | Galaxy Community[70] | https://imagej.net/ij/ |
| Galaxyproject | Schneider et al.[71] | https://usegalaxy.eu/ |
| Qualimap | Okonechnikov et al.[61] | N/A |
| FastQC | N/A | http://www.bioinformatics.babraham.ac.uk/projects/fastqc/ |
| RNA Star | Dobin et al.[60] | N/A |
| FeatureCounts | Liao et al.[62] | N/A |
| DESeq2 | Love et al.[63] | N/A |
| Pheatmap | N/A | https://cran.r-project.org/web/packages/pheatmap/pheatmap.pdf |
| Ggpubr | N/A | https://cran.r-project.org/package=ggpubr |
| Enhanced volcano plots | N/A | https://bioconductor.org/packages/release/bioc/vignettes/EnhancedVolcano/inst/doc/EnhancedVolcano.html |
| ShinyGO v0.77 | Ge et al.[64] | http://bioinformatics.sdstate.edu/go77/ |
| PhotoScapeX | N/A | http://x.photoscape.org/ |

## Article

***Continued***

| REAGENT or RESOURCE | SOURCE | IDENTIFIER |
|---|---|---|
| Other | | |
| Novex TBE-Urea Gels, 10% | Invitrogen | Cat #EC68755BOX |
| Novex TBE-Urea Gels, 6% | Invitrogen | Cat #EC6865BOX |

## EXPERIMENTAL MODEL DETAILS

### Animals

Male and female C57BL/6 mice, aged between 8 and 12 weeks, were used (Jackson Lab; Bar Harbor, ME). Mice were bred for up to 20 generations in a pathogen-free environment at the Laboratory Animal Biotechnology Unit of the Pasteur Institute of Montevideo. All experiments were performed under strict adherence to the guidelines set by the National Commission for Animal Experimentation. Experimental procedures were approved by the Ethics Committee on Laboratory Animals of the Pasteur Institute of Montevideo (protocol: 022-22). For primary cell cultures both male and female C57BL/6 mice were used. For *in vivo* experiments, C57BL/6 female mice were used.

### Cell lines

U-2 OS and MFC-7 cells were cultured in DMEM supplemented with 10% FBS. Raw264.7 cells were cultured in DMEM (high glucose) supplemented with 10% FBS, 1 mM sodium pyruvate and 100 U/mL penicillin-streptomycin. THP-1 cells were cultured in RPMI with glutaMAX supplemented with 10% FBS, non-essential amino acids, 1 mM sodium pyruvate, 10 mM HEPES and 100 U/mL Penicillin-Streptomycin. THP-1-Dual and THP-1 Dual *MAVS* knockout reporter cell lines (Invivogen, USA) were cultured in RPMI with glutaMAX, supplemented with 10% FBS, 25 mM HEPES, 2mM Glutamine, 100 U/mL Penicillin-Streptomycin and 100 μg/mL Normocin. Selection pressure was maintained with 10 μg/mL Blasticidin and 100 μg/mL Zeocin. THP-1, THP-1-Dual and THP-1 Dual *MAVS* knockout monocytes were differentiated to macrophages by 3 h stimulation with 40 nM Phorbol 12-myristate 13-acetate (PMA), followed by 48–72 h incubation without PMA.

### Primary cell culture

Differentiated Bone Marrow Derived Cells, containing dendritic cells, were cultured from bone marrow cell precursors as described in.[59] Briefly, bone marrow cells were collected form 8–12 weeks-old C57Bl/6 mice and differentiated with 0.4 ng/mL GM-CSF for 8 days in RPMI supplemented with 10% heat-inactivated FBS, 2mM glutamine, non-essential amino acids, 1 mM sodium pyruvate, 10 mM HEPES, 0.05 mM *β-mercaptoethanol* and 100 U/mL Penicillin-Streptomycin. At day 8, BMDCs were harvested and used as needed.

## METHOD DETAILS

### Total E. coli RNA purification and fractionation

Total RNA was extracted from exponentially growing *E. coli* DH5α cells with OD = 0.6. Cells were lysed in RLT buffer supplemented with 20 mM DTT using a bullet blender homogenizer (Next Advance). The homogenate was precipitated with 2.5 volumes of 100% acetone. The resulting pellet was resuspended in TRIzol, and total RNA was purified from the aqueous phase using a Monarch RNA cleanup kit. To isolate large (>200 nt) and small (<200 nt) RNAs, total RNA was sequentially bound to Monarch spin columns using binding buffer with 33% ethanol (allows binding of only >200 bp RNAs) and 66% ethanol (binds all RNAs). Additionally, 23S, 16S and 5S ribosomal RNAs were selectively removed with a NEBNext rRNA Depletion Kit for bacteria. When required, total *E. coli* RNA was degraded with recombinant human RNase 1.

### RNA-seq and bioinformatic analysis

BMDCs (300,000 cells per well) cultured in growth medium with 10% FBS and supplemented with 10 μg/mL Polymyxin B were stimulated with 100 ng/mL naked total RNA from *E. coli* or naked total self-RNA from BMDCs (isolated from cultures at day 8), in the presence of either 80 U/mL RI or heat-inactivated RI (RIΔ) for 6 h. Afterward, cells were lyzed with TRIzol and RNA present in the aqueous phase after addition of chloroform was purified with a Monarch RNA cleanup kit. RNA integrity was confirmed by electrophoresis in 10% TBE-urea polyacrylamide gels. RNA was precipitated with 0.1 volumes of 3M sodium acetate pH = 3.2 and 2 volumes of 100% ethanol and shipped to Macrogen (Korea) for paired-end mRNA sequencing (TruSeq Stranded mRNA, Illumina). Then, FastQ files containing paired-end sequencing information were mapped to the mouse genome (GRCm39) using STAR[60] (default settings). Mapping quality was assessed with QualiMap[61] (default settings). Expression of mRNAs across samples was analyzed with FeatureCounts[62] (default settings and paired-end reads counted as a single fragment) using Ensembl mouse annotation (Release v109). Differential expression across samples was computed with DESeq2[63] (default settings). Heatmaps showing topmost differentially expressed genes were calculated using rlog-transformed counts and the pheatmap R package and row-scaled. Volcano

**Cell Genomics**
Article

plots and MA-plots were constructed using EnhancedVolcano and ggpubr R packages, respectively. Differentially expressed genes between conditions of naked *E. coli* RNA with RI and naked *E. coli* RNA with inactivated RI having fold change >2, adjusted *p*-value < 1x10$^{-5}$ and base mean >16 were used for pathway enrichment analysis using ShinyGO (v0.77).[64] To determine expression differences between RNA sensors, fragments per kilobase per million (FPKM) were calculated for *Tlr3, Tlr7, Tlr8, Tlr13, Rig-I, Mda-5.* Raw sequencing data can be accessed at NCBI under BioProject: PRJNA1120732; SRA: SRP512141.

### Bioinformatic analysis of publicly available datasets

Publicly available transcriptomic datasets of THP-1 cells differentiated to THP-1 macrophages with PMA (GEO: GSE130011),[65] human monocyte-derived dendritic cells stimulated with different TLR agonists (GEO: GSE125817)[66] and Raw264.7 macrophages (GEO: GSE103958)[67] were downloaded from the NCBI SRA repository and subjected to a similar bioinformatic pipeline as previously described.

### Cell stimulation

BMDCs seeded in 24 well plates (300,000 cells/well) cultured with growth medium containing 10% FBS were stimulated for either 6 or 24 h with the following RNAs at the indicated dose: total *E coli* RNA, rRNA-depleted *E coli* RNA, size-fractionated *E. coli* RNA, synthetic RNAs Ec12 and Ec12s6G (sequences provided in the key resources table) and high molecular weight poly(I:C), together with 80 U/mL RI, thermally inactivated RI (RIΔ), or without RI. RPMI or Dulbecco's phosphate-buffered saline (DPBS) were used as negative controls and 100 ng/mL R848 was used as a positive control. Whenever bacterial RNA was used, cells were cultured with 10 μg/mL Polymyxin B to quench any endotoxins remaining after RNA purification. After 6 h, the expression of *Il1b, Il1a, Il6, Cxcl10, Ifit2,* and *Oas3* was analyzed by RT-qPCR (See primers in Table S1). Secreted TNF-α was quantified in cell-conditioned medium by ELISA. After 24 h, cell surface expression of CD40 and CD86 was analyzed by flow cytometry. To study involvement of dynamin-dependent endocytosis on internalization and responses of naked inflammatory RNAs, BMDCs were stimulated for 6 h with 1 μg/mL naked total RNA with or without 80 U/mL RI in the presence of 80μM Dynasore or vehicle (DMSO). Dynasore was preincubated for 1 h before the addition of the stimuli. In other experiments, BMDCs were stimulated with 0.5 μg/mL naked fluorescein-labeled poly(I:C) or Dextran-AF647 in the presence or absence of 120 U/mL RI for 1 h, and analyzed by confocal microscopy and flow cytometry.

Raw264.7 cells, seeded in 24 well adherent plates (at 200.000–300.000 cells/well) and cultured in 10% FBS-containing medium, were stimulated for 6 h with naked total *E coli* RNA at 1 or 2 μg/mL, naked poly(I:C) at 10 μg/mL or R848 at 1 μg/mL. When indicated, 80 U/mL RI was added. M0549, a TLR7/8 selective inhibitor, was used at 1 μM and preincubated for 3 h before stimuli. A MyD88 inhibitory peptide was used at 1 or 10 μM and preincubated for 4 h before stimuli.

THP-1 cells, seeded in 24 well adherent plates (250.000–350.000 cell/well) and cultured in 10% FBS-containing medium, were differentiated to THP-1 macrophages with 100 nM PMA for 48 h. THP-1 monocytes or THP-1 macrophages were stimulated for 22 h with naked *E. coli* RNA at 1 μg/mL, naked poly(I:C) at 10 μg/mL, or R848 at 1 μg/mL. RI was added at 80 U/mL. In other experiments, THP-1 cells, cultured in 10% FBS, were seeded in 96 well plates (100.000 cells/well) and stimulated with 0.5 μg/mL naked fluorescein-labeled poly(I:C) or Dextran-AF647 in the presence or absence of 160 U/mL RI for 180 min and analyzed by flow cytometry.

THP-1 dual wild type and *Mavs* knockout cells (Invivogen) seeded in 96 well adherent plates and cultured with growth medium containing 10% FBS were differentiated to macrophages with PMA. Then THP-1 derived macrophages were stimulated for 24 h with 5 μg/mL naked poly(I:C), with or without 80U/mL RI. RPMI and 200 ng/mL LPS were used as negative and positive controls, respectively. After 24 h, luciferase activity in the medium was detected following manufacturer's instructions using a LUMIstar Optima luminometer.

### RT-qPCR

BMDC RNA was extracted with TRIzol following manufacturer's instructions and quantified spectrophotometrically. cDNA was synthesized from 300 to 1000 ng total RNA using M-MLV reverse transcriptase (Thermo). Briefly, total RNA was treated with 2 units of DNase I for 20 min at 25°C to eliminate genomic DNA, in a reaction volume of 10 μL. DNase I was inactivated by addition of EDTA at a final concentration of 4.5 mM and heating at 75°C for 10 min. Then, RNA was reverse-transcribed in a 20-μL reaction volume using oligo(dT)$_{18}$ primers, following manufacturer's instructions and employing a two-step PCR program: 52 min at 37°C, followed by 15 min at 70°C. Then, cDNA was diluted 1/5 and stored at −20°C until use. Gene expression was quantified by qPCR in a QuantStudio 3 Real-Time PCR System (Applied Biosystems) employing a FastStart Universal SYBR Green Master Mix. Briefly, a 10-μL reaction was carried out using 2 μL of cDNA, a primer mix at 0.3 μM final concentration, and 5 μL of an SYBR Green Master Mix. The qPCR program was: 2 min at 50°C, followed by 10 min at 95°C, followed by 40 cycles of 15 s at 95°C and 1 min annealing/extension at 60°C–65°C, and a final melt curve stage. Comparative gene expression analysis was calculated using the 2$^{-ΔΔCT}$ method. Data was normalized against mock treatment (RPMI or DPBS) and *actb* was used as a housekeeping, reference gene.

### Tlr13 knockdown in Raw264.7

*Tlr13* knockdown was carried out in Raw264.7 cells with DsiRNAs (IDT) and RNAiMAX as transfecting agent. 80.000 cells seeded in 24 well adherent plates were cultured in complete growth medium for 24 h. Afterward, fresh growth medium (450 μL; without

antibiotics) containing 50 μL of preformed DsiRNA-lipid complexes was added to each well. Cells were incubated for 24 h with DsiRNA-lipid complexes containing either control DsiRNA or an equimolar mixture of two DsiRNAs targeting TLR13. Afterward, the medium was replaced for 1,000 μL of fresh growth medium containing antibiotics and cells were cultured for additional 24 h. To assess *Tlr13* silencing, cells were lyzed with TRIzol and *Tlr13* was measured by RT-qPCR. To assess the functional involvement of TLR13 in the recognition of naked exRNAs, cells were stimulated with naked *E. coli* RNA for 6 h in the presence of 80 U/ml RI and 20 μg/mL Polymyxin B. The levels of *Il6* were measured by RT-qPCR. DsiRNA-lipid complexes were made using lipofectamine RNAiMAX according to manufacturer's instructions. Briefly, a ratio of 2.5 μL of DsiRNA at 10 μM, 0.8 μL RNAiMAX, and 50 μL Opti-MEM (Thermo), was used for each well. DsiRNAs were used at a final concertation of 50 nM. Using a fluorescent DsiRNA-tye563, we confirmed transfection efficiencies of 80%, and minimal cell death.

### In vitro transcription

mRNAs coding for *nanoLuc* or *eGFP* were synthesized by *in vitro* transcription: nanoLuc and eGFP coding sequences were amplified by PCR from the pUAS-NanoLuc[68] and pEGFP-N2 plasmids, using primers shown in Table S2. Forward and reverse primers were designed to incorporate a T7 RNA polymerase sequence and a Kosak sequence, and a 30–31 nt polyA tail, respectively. PCR reactions were carried out with Phusion Polymerase. The amplification program was: 30 s at 98°C, followed by 30–35 amplification cycles, each comprising 10 s at 98°C, 10 s at 60°C, and 30 s at 72°C. A final extension step of 10 min at 72°C was included. PCR products were analyzed by 1% agarose gel electrophoresis and purified with PureLink quick gel extraction kit directly from the PCR solution (for *nanoLuc*) or from excised gel bands (for *eGFP*). DNA was concentrated by overnight ethanol precipitation and used as template for mRNA synthesis. *nanoLuc* and *eGFP* mRNAs were synthesized and co-transcriptionally capped using HiScribe T7 High Yield RNA Synthesis Kit. Briefly, reactions were performed in a 20-μL reaction volume at 37°C for 2 h using 1 μg of template and a 3′-O-Me-m7G (5′)ppp(5′)G RNA Cap Structure Analog, with 4:1 CAP:GTP ratio, and following manufacturer's instructions. For *eGFP*, UTP was substituted with N1-Methylpseudouridine-5′-Triphosphate. Afterward, DNA templates were digested with Turbo DNase I. In vitro-synthesized mRNAs were purified with a Monarch RNA Cleanup Kit and stored at −80°C until use.

### In vitro translation

*NanoLuc* mRNA was translated *in vitro* with Retic Lysate IVT Kit following manufacturer's instructions. Briefly, 100 ng of *NanoLuc* mRNA was incubated for 45 min at 30°C with a rabbit reticulocyte extract. Afterward, nanoLuc bioluminescence was measured using a Bio-Glo-NL Luciferase Assay System and detected in a LUMIstar Optima luminometer.

### RNase activity in mouse serum and peritoneal wash

Blood from three female mice was collected from the submandibular vein and allowed to clot for 45 min at room temperature. Serum was obtained by centrifuging the blood for 15 min at 4°C. A cell-free peritoneal wash was collected from the same mice. To that end, 3.3 mL DPBS was introduced into the peritoneal cavity, collecting 2.5 mL of peritoneal wash fluid. To remove peritoneal cells and debris, the peritoneal wash was centrifuged four times: 5 min at 300 g, 6 min at 450 g, 6 min at 500 g, and 10 min at 2,000 g. RNase activity in both serum and peritoneal wash was measured using a continuous fluorometric assay with the RNaseAlert Lab Test kit (Thermo). Briefly, 5 μL serum (diluted 1/100 in DPBS) and 5 μL of an identical effective dilution of peritoneal wash (1/3 in DPBS; assuming a peritoneal volume of 100 μL[69] that was previously diluted to 3.4 mL with DPBS) were assayed, following the manufacturer's instructions. Fluorescence was measured using a Varioskan fluorimeter for over 35 min, with excitation at 490 nm and emission at 520 nm. RNase activity was determined based on the initial reaction velocities and expressed as a percentage relative to the sample with the highest RNase activity.

### nanoLuc mRNA internalization and translation assays

Naked nanoLuc mRNA capture and translation was assessed in murine primary cells (BMDCs) and two human epithelial cell lines (U-2 OS and MCF-7). To quantify *nanoLuc* mRNA levels and nanoLuc protein in BMDCs (Figures 4B and 4C), cells were seeded in 24 (250.000 cells/well) or 48 (120,000 cells/well) well plates and cultured for 2 h to allow adhesion. Then, growth medium containing FBS was removed, cells were washed, incubated in growth medium without FBS and stimulated for 2 h with 1 μg/mL naked *nanoLuc* mRNA with 80 U/mL RI, thermally inactivatedRI, or without RI. To investigate the impact of varying concentrations of FBS and RI on *nanoLuc* translation by BMDCs (Figure 4D), cells were seeded in 48 well plates (120,000 cells/well) and cultured for 2 h to allow adhesion. Then, growth medium containing FBS was removed, cells were washed, incubated in growth medium containing varying concentration of FBS (0, 0.1, 1, 10%) and RI (0, 80, 120, 240, 320 U/mL) and stimulated with 1 μg/mL naked *nanoLuc* mRNA for 2 h. Subsequently, the medium containing the stimuli was removed and fresh growth medium with 10% FBS was added. After 6 h, *nanoLuc* mRNA was quantified by RT-qPCR. The RT-qPCR protocol was specifically designed to enable detection of full-length, undegraded *nanoLuc* mRNA. This was achieved by designing primers that selectively amplified the transcript 5′ end, which had been previously reverse-transcribed using oligo(dT) primers. After 24 h, nanoLuc protein translation in BMDCs was detected with Bio-Glo-NL Luciferase Assay System using a LUMIstar Optima luminometer. MCF-7 and U-2 OS cells were seeded in 48 well adherent plates. After reaching 80% confluence, growth medium was replaced with growth medium containing 0.1% FBS. Then, cells were incubated with varying concentration of *nanoLuc* mRNA (0, 0.25, 1, 4μg/mL), with either 80 U/mL RI or thermally inactivated RI for 2 h. To assess the involvement of dynamin-dependent endocytosis in naked *nanoLuc* mRNA capture, U-2 OS cells were incubated for 2 h with

80 μM Dynasore or DMSO, before addition of 1 μg/mL *nanoLuc* mRNA, with or without 80 U/mL RI. After 2 h, media was replaced with growth medium containing 10% FBS. Cells were cultured for additional 24 h, and nanoLuc protein levels were measured as previously described.

### Naked eGFP internalization and translation assays

U-2 OS cells were seeded onto 35-mm glass bottom dishes (100,000 cells) and incubated overnight with growth medium. The next day, cells were washed 3 times with MEGM (Lonza), and MEGM either without or with 0.1% FBS was added. Cells were incubated for 2 h with 5 μg/mL naked eGFP mRNA in the presence or absence of 80 U/mL RI, or with a DOTAP-eGFP mRNA complex. After the incubation, cells were washed 3 times and cultured for an additional 24 h in complete growth medium, containing 10% FBS. To visualize and analyze eGFP translation, cells were imaged using a Zeiss LSM 880 confocal microscope. Nuclei were stained with NucRed (Thermo). Imaging was performed utilizing a 63× oil immersion objective with a numerical aperture of 1.4, employing a Plan Apochromat configuration. The imaging acquisition parameters were set with a pixel dwell time of 4.10 μs and size of 0.55 microns, utilizing bidirectional scanning. The total scanning duration for each acquisition was 1 min and 20 s, with an image size of 655.8 x 655.8 microns. Images were subsequently processed and analyzed using ImageJ. The segmentation was conducted on both the green (eGFP) and red (nuclei) fluorescence channels. Quantitative analysis of the cellular responses was executed through Integrated Density calculations. Prior to calculations, a noise subtraction step was implemented to ensure data integrity. The resultant processed images were subjected to total image calculations, facilitating the extraction of Integrated Density values indicative of cellular responses. The image in the GFP channel (488 nm) was processed by setting the minimum value at 3 and the maximum value at 8. Then, a Gaussian blur filter of value 2 was applied. The same was done for data visualization and figure creation. Images were cropped to aid visualization.

### *In vivo* experiments

For intraperitoneal administration of RNAs, female mice were *i.p* inoculated with 250 μL of varying doses (0.025–62.5 μg) of naked total *E. coli* RNA with or without 40, 200 or 400 units of RI in DPBS. DPBS and 62.5 μg LPS were used as positive and negative controls, respectively. After 24 h, animals were euthanized and 4–5 mL of DPBS supplemented with 3 mM EDTA was injected into the peritoneal cavity to extract peritoneal cells. Peritoneal cells were pelleted at 375 g for 5 min at 4°C, resuspended in FACS buffer (0.5% w/v BSA, 2 mM EDTA in DPBS), counted, and stained for flow cytometry analysis. For systemic administration of RNAs, female mice were retro-orbitally inoculated with 100 μL of either high molecular weight naked poly(I:C) (200 ng or 1000 ng) or naked total *E. coli* RNA 10 μg in the presence or absence of 480 units of RI in DPBS. DPBS with or without RI was used as control. After 6 h, mice were euthanized, and spleens were collected. To obtain spleen cells, spleens were incubated in 1mL of collagenase solution (2 mg/mL Collagenase D, 2% v/v FBS in RPMI), cut on ice with a bistoury and incubated for 20 min at 37°C. The reaction was stopped with 100 μL of 100 mM EDTA. Spleen cells were passed through a 100 μm filter and resuspended in 40 mL of PES buffer (0.7 mM EDTA, 2% v/v FBS in DPBS). Cells were pelleted at 1500 rpm for 15 min at 4°C, resuspended in 5 mL RBCL buffer (154 mM NH$_4$Cl, 10 mM KHCO$_3$, 0.1 mM EDTA; pH = 7.2–7.4), and incubated for 10 min at room temperature to lyse red blood cells. Then, spleen cells were washed two times with PES solution, resuspended in FACS buffer, counted, and stained for flow cytometry analysis.

### Flow cytometry

Cells suspended in FACs buffer were seeded in V-shaped 96 well plates (200,000–500,000 cells/well). When live/dead viability dye was used, cells were stained with 200 μL of 1/200 live/dead violet in FACS buffer and incubated on ice for 30 min in the dark. If viability was assessed with DAPI, the previous step was not performed. Then, cells were pelleted at 2200 rpm for 2 min and blocked with 25 μL of 10% normal rat serum in FACS buffer for 30 min on ice. For cell surface antibody staining, 25 μL of 2x antibody mix (see Table S3) in FACS buffer was added, and cells were stained for 30 min on ice in the dark. Stained cells were washed 2 times with FACS buffer, resuspended in 200 μL FACS buffer, and analyzed by flow cytometry. To assess viability, 0.5 μg/mL final concentration of DAPI was added immediately before sample analysis. Myeloid and lymphoid spleen cells were analyzed in a Cytek Aurora spectral flow cytometer. Peritoneal immune cells were analyzed in an Attune NxT flow cytometer or in a Cytek Aurora spectral flow cytometer. BMDCs cells were analyzed in a BD Accuri C6 flow cytometer or in a Cytek Aurora spectral flow cytometer.

### RNA polyacrylamide electrophoresis

RNAs were analyzed by polyacrylamide gel electrophoresis under denaturing conditions using 6–10% acrylamide TBE-urea gels. Briefly, RNA samples were mixed with 2x RNA loading dye, heated at 65°C for 5 min, cooled on ice and analyzed on an XCell *SureLoc* Mini-Cell Electrophoresis system (Invitrogen) using 0.5x TBE as running buffer. Gels were run at constant voltage (160V) for 60–80 min. Gels were stained for 15 min with 1/10000 SYBR Gold in 0.5x TBE and imaged with a transilluminator (Loccus biotecnologia) with a coupled imaging platform (Carestream). When needed, images were cut and contrasted using PhotoScapeX.

## QUANTIFICATION AND STATISTICAL ANALYSIS

Statistical analysis was performed using R or GraphPad Prism 9. For RNA-seq experiments, 3 biological replicates for all 4 experimental conditions were performed. Significant differences in gene expression across samples was computed with DESeq2 and multiple hypothesis testing was controlled for using the Benjamini-Hochberg method to estimate false discovery rates (FDR). Differentially expressed genes with adjusted $p$p-values lower than 0.001 were reported. For *in vitro* experiments, each dot in bar plots represents a biological replicate. For *in vivo* experiments, each dot in bar plots represents a different mouse. Bar plots correspond to mean, error bars to standard deviation. For microscopy experiments, each dot in violin plots represents an individual cell. When statistical analysis was performed, normal distribution of data was assessed using Shapiro-Wilk and Kolmogorov-Smirnov tests. For statistical comparisons between two conditions, unpaired two-tailed t-test was used when the data followed a normal distribution, and unpaired two-tailed Mann-Whitney test was applied when the data did not follow a normal distribution. For multiple comparisons, if data was normally distributed and had variance homogeneity, one-way ANOVA was used. If data was normally distributed but had no variance homogeneity, Welch ANOVA test was used. If data had no normal distribution, Kruskal-Wallis test was used. Mouse serum and peritoneal wash RNase activity was compared using a paired two-tailed t test. Statistically significant differences are indicated with asterisks (*, p-value < 0.05, **: p-value < 0.01, ***: p-value < 0.001, ****: p-value < 0.0001).

