## [Document S2. Transparent peer review records for Castellano et al. · Cell Genomics]

Summary

Initial submission: Received: 4/27/2024

Scientific editor: Laura Zahn

First round of review: Number of reviewers: 2
Revision invited: 7/25/2024
Revision received: 2/25/25

Second round of review: Number of reviewers: 1
Accepted: 4/10/25

Data freely available: Yes

Code freely available: N/A

This transparent peer review record is not systematically proofread, type-set, or edited. Special characters, formatting, and equations may fail to render properly. Standard procedural text within the editor's letters has been deleted for the sake of brevity, but all official correspondence specific to the manuscript has been preserved.

Referees' reports, first round of review

Reviewer #1: In the excellent study, the authors have demonstrated that naked RNA is bioactive and does not need encapsulation inside synthetic or biological lipid vesicles for functional uptake, highlighting the importance of nonvesicular extracellular RNA-mediated intercellular communication. The study is generally well designed and performed. It could be accepted after the following questions being resolved.

- (1) It is important to test whether the RNA forms complex (nanoparticles) with certain proteins or other biomolecules, and thus endocytosed by the cells. Identification of the complex components is suggested.
- (2) It is also interesting to know whether RNA is uptake at similar efficiency as DNA.

Reviewer #2: In this manuscript, the authors determine that addition of RNase inhibitors to cell culture supernatants is sufficient to significantly enhance the activity of naked extracellular RNAs on target cells. The authors provide evidence that these naked RNAs may both interact with cell surface receptors as well as gain access to the cytosol of target cells to have immunostimulatory functions. There are several key strengths of this manuscript. First it addresses the very important question of whether and how RNAs found in the extracellular space might impact target cell function, ultimately allowing these RNAs to communicate signals between cells. It provides several lines of convincing data that addition of RNase inhibitors to cell culture supernatants can enhance the transfer and functions of naked RNAs to target cells in vitro. It uses primary cells (BMDC) to test for response in many of the experiments, which provide a physiologically relevant cell type. Experiments generally include good controls. And the study uses sound methodologies and provides sufficient details for the scientific community to support the rigor and reproducibility of this work.

However, there are several significant limitations. First, throughout the manuscript, the authors switch between using whole bacterial RNAs and polyIC (a synthetic dsRNA analog that classically is thought to mimic RNA products seen during viral infection). It is not clear whether these RNAs should or do act equivalently, given the very different biology and structure of these immunostimulatory RNA sources. There are also several conclusions that are suggested, but incompletely supported, by experiments including that TLRs are sensing RNase-protected naked RNAs and that the RNAs taken up into cells are located within endosomes. Although in vivo data support that endogenous RNase activity may be important in limiting naked RNA transfer in some biofluids, the effects are not consistent between models and the cellular responses do not always correlate directly with the in vitro results, limiting the ability of the reader to understand the significance of the manuscript's findings.

Major Points:

1. In Figure 2A, a single mRNA is quantified to test the effects of an endocytosis inhibitor on BMDC exposed to naked RNAs with or without RNase inhibitor. This result should be supported by looking at additional changes observed (by RNA and/or protein) to test the robustness of the result.
2. In Figure 2C, the authors conclude that observing puncta of polyIC in cells "positions internalized naked RNA ... inside endosomes" (line 106). This is not sufficient evidence to pinpoint sub-cellular localization. Also, does bacterial RNA + RNase inhibitor show a similar localization? If not, how would this change the conclusions of the paper?
3. In Figure 3, only indirect evidence is presented for the involvement of TLR13 in recognition of naked bacterial RNA. Since this is a major claim of the paper (i.e. abstract "In murine cells, this response mainly depends on the action of endosomal Toll-like receptors."), direct evidence is needed to support this claim.
4. Are human cells unable to respond to naked bacterial RNA + RNase inhibitor if they lack TLR13?
5. Is an intracellular sensing pathway (i.e. MAVS, Figure 4) relevant only for cells that lack TLRs to sense endosomal RNAs?
6. If both mouse peritoneal wash and mouse serum have relatively poor RNase activity when compared with FBS (Fig 7A), then should naked RNA transfer be readily detected in vivo in murine models?
7. Why are there only changes in IV polyIC induced, but not bacterial induced, immune responses with the addition RNase inhibitor (Fig 7F, G)? Would you not expect that the more stable polyIC dsRNA analog be

less sensitive to an incremental loss of low-level RNase activity in the mouse serum?

8. How do you connect the naked RNA in vitro effects on DC/Macs observed to the changes in adaptive immune cells observed in vitro? Do you see direct effects of RNA on the activation state of lymphocytes in vitro?

Minor Points:

1. The claim that the "main mechanism by which RI facilitates naked exRNA uptake is by stabilizing RNA" could be strengthened with titration studies to quantify what level of RNA is required to see effects in target cells.

2. Is there a reason that an osteosarcoma cell line is used in Figure 5H/I? This seems quite biologically distinct from RNAs targeting antigen presenting cells (i.e. DC and macrophages which are used in most of the other models).

3. Although it is possible that enhanced RNase activity in serum could protect from severe systemic inflammation (lines 205-206), there is no direct evidence to support this conclusion in the paper. It might be better to include this concept in the discussion rather than results section. How the balance of RNA sensing serves to benefit immune responses vs damage hosts is an important question that will require careful future work, requiring consideration of both the beneficial and detrimental effects of inflammation (lines 279-283).

4. In Supp Fig 4B, what is the major population of cells that you gate out on FWD-SSC?

5. There are several places where imaging and flow experiments are not quantified over multiple replicates. This would help the reader have confidence in the robustness of the result ... ie Fig 2B, 2E, 5I, 6B, S6.

Authors' response to the first round of review

Dear Editor,

We would like to thank both reviewers for their excellent questions. We think we have experimentally addressed all their concerns, further reinforcing the conclusions of this paper (that both reviewers considered important). The main additions to the earlier version of the manuscript are:

- The addition of a murine macrophage cell line (Raw264.7) and a murine dendritic cell line (JAWSII) to complement experiments done in murine BMDCs and human THP-1 cells.
- The addition of drugs to inhibit TLR7/8 and MyD88 in Raw264.7, to further support the involvement of TLR13 in the recognition of naked bacterial extracellular rRNA.
- The knockdown of TLR13 in Raw267.7 cells and its effect on the recognition of naked bacterial exRNAs.
- The quantification of naked RNA endocytosis rates in various models. In addition to the identification of an RNA sequence that is inherently stable and works as a TLR13 agonist.
- The addition of a new experiment in BMDCs that acts as a conceptual link between our in vitro assays and our in vivo assays (comparison of the RI effect when cells are grown in the presence or absence of FBS).
- The colocalization between internalized RNA and RAB5 to show the transient presence of internalized RNA in endosomes.
- The inclusion of poly(I:C) in i.p. injection experiments (now poly(I:C) and E.coli RNA tested in both i.p. and i.v. injections).
- The addition of more biological replicates (i.e., mice) to strengthen in vivo data.
- The addition of replicates in several in vitro assays that required statistics.
- Changes throughout the text, especially in the results section and the discussion.

Reviewers' Comments:

Reviewer #1:

In the excellent study, the authors have demonstrated that naked RNA is bioactive and does not need encapsulation inside synthetic or biological lipid vesicles for functional uptake, highlighting the importance of nonvesicular extracellular RNA-mediated intercellular communication. The study is generally well designed and performed. It could be accepted after the following questions being resolved.

- (1) It is important to test whether the RNA forms complex (nanoparticles) with certain proteins or other biomolecules, and thus endocytosed by the cells. Identification of the complex components is suggested.**

The reviewer is correct in that, although we spiked naked RNAs into the media / biofluid, these could form a complex with other soluble factors present in the media that could facilitate RNA uptake. To study this, we repeated some of our main assays in serum-deprived media after thoroughly washing the cell monolayer, to remove most cell-secreted factors and proteins present in FBS. Surprisingly, the cell response to naked RNA was comparable to that obtained when the experiments were performed in the presence of serum and without medium renewal. These results (left) are now shown in Figure 1H.

In a second set of experiments, we added labeled RNAs in fresh serum-free media for as short as 5 minutes and then removed the labeled RNA. Even under these very stringent conditions, we were able to detect the mark in the cells by confocal microscopy. These results are part of a second manuscript we are preparing that focuses on the subcellular mechanisms of naked RNA endocytosis and endosomal escape. We would prefer not to add this data in the final version of the manuscript, as the cell model and the identity of the labeled RNAs are different to those used herein. However, considering the new Figure 1H and these “short pulse” results,

we think it is unlikely that other soluble factors are needed for efficient gymnotic uptake. However, we also added the following phrase in the discussion to contemplate the sharp point raised by the reviewer:

It should be noted that we used the term “naked RNA” throughout this study because all stimuli were added in the absence of proteins or transfection reagents, including lipid nanoparticles. However, we cannot rule out the possibility that these naked RNAs might form a complex with one or more soluble extracellular protein(s) or other binding partners before cellular uptake. Having said that, the fact that similar responses were obtained in experiments performed in FBS-containing and FBS-free fresh media (Figure 1H) argues against this possibility.

(2) It is also interesting to know whether RNA is uptake at similar efficiency as DNA.

To test this, we compared the uptake of the same oligonucleotide (Ec12), either in the context of a double-stranded DNA molecule (i.e., two complementary DNA oligonucleotides were annealed together, only one being labeled in 5' with Cy5) or a single-stranded RNA molecule. We chose the sequence of the Ec12 RNA (new **Figure 2G**) because it lacks uridines and has a low pyrimidine content, and hence its stability is expected to be high irrespectively of the nature of the backbone, even when incubated as a single-stranded RNA (see our response to reviewer 2, question 7). Ten-fold serial dilutions from 1000 to 0.01 nM showed that both oligonucleotides were internalized by 100% of THP-1 cells when incubated in FBS-containing medium without the need of RI (new **Figure S5**, see data below). In addition, EC50 values were almost identical for the ssRNA and the dsDNA. This strongly suggests that the presence of a 2'-OH or a 2'-H in the ribose, or the double-strand vs single-strand nature of the molecule does not appear to play a major role in its internalization, at least for short oligonucleotides.

Of note, Ec12 ssRNA uptake efficiency (100%) sharply contrasts with that observed for the naked dsRNA, poly(I:C)-FITC (between 8-12%, new **Figure 3F and S7A-B**). However, this lower efficiency can be explained by several reasons: larger size, higher pyrimidine content (note the uptake is, in this case, RI-dependent), and a different fluorophore. For all these reasons, we cannot yet ascertain that RNA size correlates with uptake efficiency, although this would be reasonable.

In addition, these results encouraged us to add a new paragraph in the discussion, where we suggest that TLR13 might have evolved to sense the Ec12 sequence due to its extracellular stability and not only because of its conservation in bacteria. We would like to thank the reviewer for this question that encouraged us to do these experiments.

Reviewer #2:

In this manuscript, the authors determine that addition of RNase inhibitors to cell culture supernatants is sufficient to significantly enhance the activity of naked extracellular RNAs on target cells. The authors provide evidence that these naked RNAs may both interact with cell surface receptors as well as gain access to the cytosol of target cells to have immunostimulatory functions. There are several key strengths of this manuscript. First it addresses the very important question of whether and how RNAs found in the extracellular space might impact target cell function, ultimately allowing these RNAs to communicate signals between cells. It provides several lines of convincing data that addition of RNase inhibitors to cell culture supernatants can enhance the transfer and functions of naked RNAs to target cells in vitro. It uses primary cells (BMDC) to test for response in many of the experiments, which provide a physiologically relevant cell type. Experiments generally include good controls. And the study uses sound methodologies and provides sufficient details for the scientific community to support the rigor and reproducibility of this work.

However, there are several significant limitations. First, throughout the manuscript, the authors switch between using whole bacterial RNAs and polyIC (a synthetic dsRNA analog that classically is thought to mimic RNA products seen during viral infection). It is not clear whether these RNAs should or do act equivalently, given the very different biology and structure of these immunostimulatory RNA sources.

We agree that, in the original version of the manuscript, not all experiments were done with different stimuli in a systematic manner. Now this has been improved: the two in vivo models were tested with both RNA types, and in the presence or absence of RI. In the peritoneal cavity, a dose-response study was also performed for both stimuli. We also expanded our in vitro work and tested both stimuli in our main in vitro models: BMDCs and macrophages (THP-1 and now also Raw264.7). New data is found throughout the manuscript, especially in the **new figures 2, 3, 5, 6 and S4**.

Thanks to this suggestion, we now have a clearer picture of the different responses against different naked RNAs, elicited by different cells due to their particular set of RNA sensors. This was combined with genetic and pharmacological perturbation of these sensors in certain cell types (see below), making this study a valuable resource for researchers interested in RNA immunology.

There are also several conclusions that are suggested, but incompletely supported, by experiments including that TLRs are sensing RNase-protected naked RNAs and that the RNAs taken up into cells are located within endosomes.

We agree with the reviewer that the earlier version of the manuscript was not particularly strong in these two aspects. We are happy to report that we have now included several new experiments showing the endosomal localization of naked exRNAs (**new Figure 3H and S7E**) and their sensing by endosomal TLRs. In particular, based on a combination of MyD88 and TLR7/8 inhibitors in cells only expressing TLR7 and TLR13, and the knockdown of TLR13 in these cells, we can

now conclude that TLR13 is the main sensor of naked bacterial RNA in murine immune cells (**new Figure 2H-L**). This is in strong agreement with our earlier results with the Ec12 synthetic RNA:

Experiments in Raw264.7

Although *in vivo* data support that endogenous RNase activity may be important in limiting naked RNA transfer in some biofluids, the effects are not consistent between models and the cellular responses **do** not always correlate directly with the *in vitro* results, limiting the ability of the reader to understand the significance of the manuscript's findings.

Thanks to this comment, we addressed the missing links between our *in vitro* and *in vivo* data. We measured the BMDC response to naked *E. coli* RNA either in media containing 10% FBS or

lacking FBS (hence, in the absence of extracellular RNases) (**new Figure 1H**; also see our first response to reviewer 1). This experiment showed that RI addition is essential for RNA-induced inflammation in FBS containing media but completely dispensable in FBS free media. This behavior recapitulates what we had seen in vivo: naked exRNAs (now both poly(I:C) and *E. coli* RNA) are inherently proinflammatory in the murine peritoneal cavity, that has 4.5-fold lower RNase activity than mouse serum (**see new Figure 6B**).

Thus, both in vitro and in vivo experiments consistently show that the need for RI negatively correlates with the RNase content in the environment in which the cells are present.

Major Points:

1. In Figure 2A, a single mRNA is quantified to test the effects of an endocytosis inhibitor on BMDC exposed to naked RNAs with or without RNase inhibitor. This result should be supported by looking at additional changes observed (by RNA and/or protein) to test the robustness of the result.

Following the reviewer's suggestion, we tested the effects of endocytosis inhibition in BMDC exposed to naked RNAs with or without RI across multiple genes including proinflammatory cytokines and ISGs. Our results clearly and robustly show that endocytosis is essential for naked RNA sensing (**new Figure 2A**).

2. In Figure 2C, the authors conclude that observing puncta of poly(I:C) in cells "positions internalized naked RNA ... inside endosomes" (line 106). This is not sufficient evidence to pinpoint sub-cellular localization. Also, does bacterial RNA + RNase inhibitor show a similar localization? If not, how would this change the conclusions of the paper?

As answered earlier, we now show the colocalization between naked RNA (poly(I:C)-FITC) and the endosomal marker RAB5 (**new figure 3H and S7E**). In addition, we also have data showing

the colocalization of short single stranded RNAs with RAB5, RAB7 and LAMP1. These data are part of a new manuscript in preparation that focuses on the mechanism of naked exRNA endosomal escape and can be shared upon request. This data confirms that endosomal localization of naked exRNA is not specific for poly(I:C). However, because this data was obtained with a different cell model than that used in this study, we would prefer not to include it in this opportunity. We think the data shown in Figure 2A, 3H and S7E already substantiates the point that naked exRNAs are internalized by endocytosis, where they can engage with endosomal TLRs.

3. In Figure 3, only indirect evidence is presented for the involvement of TLR13 in recognition of naked bacterial RNA. Since this is a major claim of the paper (i.e. abstract "In murine cells, this response mainly depends on the action of endosomal Toll-like receptors."), direct evidence is needed to support this claim.

We agree that this was an important aspect of the paper that required further experiments, and, as answered above, we now provide conclusive evidence (**new Figure 2H-L**) that naked bacterial exRNA is sensed at the endosomal level by TLR13 (at least in murine Raw264.7 macrophages where the siRNA experiments were performed). We tried to also knockdown TLR13 in BMDCs, but unfortunately all our attempts were unsuccessful. Nevertheless, we think that, in combination with our previous Ec12 and rRNA depletion results, the new experiments strongly support the claim that naked exRNAs can be sensed at the endosomal level.

4. Are human cells unable to respond to naked bacterial RNA + RNase inhibitor if they lack TLR13?

We have observed that the response of human THP-1 cells (which lack TLR13) to naked *E. coli* RNA, with or without RI, is very low compared to that seen in BMDCs and Raw264.7 cells (**new Figure S4G-I**). However, we acknowledge that the lack of evidence for naked *E. coli* RNA sensing in THP-1 cells is not equivalent to evidence for lack of sensing. For example, we cannot assure that we are using the best possible readout for the assay. Having said that, our results are in strong agreement with previous reports that show transfected bacterial RNA in either THP-1 monocytes or THP-1 derived macrophages triggers no inflammatory response, unless mouse TLR13 is ectopically expressed (Fieber et al., 2015; PMID: 25756897).

Experiments in THP-1 monocytes

We have included the following phrase in the results section:

Of note, in agreement with previous observations (Fieber et al., 2015), the human monocytic cell line THP-1, that lacks TLR13, did not respond (Figure S4G-I).

5. Is an intracellular sensing pathway (i.e. MAVS, Figure 4) relevant only for cells that lack TLRs to sense endosomal RNAs?

This is an excellent question that we have also been wondering about for a relatively long time. It turns out that when we analyze bulk RNA-seq data of different cell lines we consistently see the following pattern: cells express high levels of either endosomal TLRs or cytosolic RLRs, but usually not both. Take for example Raw264.7 that only expresses TLR7 and TLR13 (**new Figure 2H**) and THP-1 monocytes and macrophages that mainly express MAVS-associated RLRs (**new Figure 3E**). BMDCs could be considered an exception (**new Figure 3D**) since they express low levels of RIG-I and MDA5 in the context of a high expression of TLR7, 8 and 13. However, it is possible that different subpopulations in this primary cell culture explain this (Helft et al., 2015; PMID: 26084029). In any case, we did not find an evident way of testing the hypothesis on the mutual exclusiveness of endosomal vs cytosolic RNA sensing strategies. This avenue of research certainly merits further exploration, but such research would extend beyond the scope of the present manuscript.

6. If both mouse peritoneal wash and mouse serum have relatively poor RNase activity when compared with FBS (Fig 7A), then should naked RNA transfer be readily detected in vivo in murine models?

In the earlier version of the manuscript we did not use a very sensitive method to measure the RNase activity in mouse serum and peritoneal fluid. Now, using a highly sensitive fluorometric assay consisting of a short RNA sequence flanked by a fluorophore and its quencher, we found that mouse serum has a much higher (4.3-fold in average, n=3 mice) RNase activity than an

equivalent dilution of a peritoneal wash (**new Figure 6B**). Thus, the RNase activity of different biofluids follows this trend: FBS >> mouse serum > mouse peritoneal fluid.

In other words, although it is still true that mouse serum has a lower RNase activity compared to FBS, it can still degrade naked RNA relatively quickly. This is the reason why we observed a dependency on RI for RNA-induced systemic inflammation after intravenous administration of naked RNA. Based on these results, we already considered in our discussion the possibility of intercellular communication mediated by naked exRNAs in RNase-poor environment such as internal tissues and the peritoneal cavity. Our lab studies RNA-mediated intercellular communication and therefore this is something that we would like to show. However, we still do not have experimental evidence to include data supporting this possibility in this manuscript.

7. Why are there only changes in IV poly(I:C) induced, but not bacterial induced, immune responses with the addition RNase inhibitor (Fig 7F, G)? Would you not expect that the more stable poly(I:C) dsRNA analog be less sensitive to an incremental loss of low-level RNase activity in the mouse serum?

To answer this question, we have increased the number of mice in the i.v. injection experiments (both with naked *E. coli* RNA and naked Poly(I:C)) to n = 9 and n = 8, respectively. Now the statistics are stronger, and we do see a statistically significant effect of RI in the case of *E. coli* RNA (**new Figure 5**), and a more significant RI-dependent response in the case of poly(I:C).

Thus, the new data confirmed earlier results. And, while there is a dependency on RI for both stimuli, we agree with the reviewer that this dependency is unexpectedly stronger for poly(I:C). We can think in at least two possible explanations for this. First, what is actually sensed by TLR13 is not intact *E. coli* RNA, and not even intact bacterial rRNA. In contrast, both the literature (Oldenburg et., 2021, PMID: 22821982) and our data (**new Figure 2G**) show that a short sequence motif in bacterial 23S rRNA (i.e., the Ec12 sequence) is necessary and sufficient for TLR13 activation. Interestingly, this sequence lacks uridine residues and has a very low pyrimidine content, potentially making it RNase resistant. In agreement with this prediction, we demonstrated that this sequence is, indeed, inherently stable (**new Figure S5**; also see our response to reviewer 1, question 2).

From an evolutionary perspective, if TLR13 evolved to recognize a specific motif in extracellular bacterial RNAs, it would make sense for such a motif to be both conserved and stable. We added a new paragraph in the discussion presenting this idea. Second, it must be considered that recognition of exRNA molecules by immune cells in an in vivo context may depend on additional factors than RNA stability alone, such as: relative endocytosis rates, levels of RNA sensors in endosomal and cytosolic compartments and even cytokines secreted by other immune cell types beyond T and B lymphocytes that could have previously recognized these exRNAs.

8. How do you connect the naked RNA in vitro effects on DC/Macs observed to the changes in adaptive immune cells observed in vitro? Do you see direct effects of RNA on the activation state of lymphocytes in vitro?

We observed RI-dependent activation of both macrophages and plasmacytoid dendritic cells in the spleen of mice exposed to naked exRNAs (**new Figure S9**). The results are not as clean as those obtained in T and B cells, possibly due to the low number of myeloid cells in the spleen. However, the fact that these myeloid cells did respond to the stimuli suggests that they could be releasing cytokines or other ligands and soluble factors that could aid in the observed expression of early activation makers in splenic lymphocytes. In addition, lymphocytes, especially B cells, could also directly recognize the exRNAs thanks to these cells also harboring innate immune RNA sensors, including TLR7 (Hwang et al. 2012, PMID: 23150717). There are also reports showing expression of RNA-sensing TLRs in T cells (reviewed in Kaelitz, 2007; PMID: 17129718).

We added the following phrase in the results section:

These results are in good agreement with previous findings using LNP-formulated RNAs³⁰. We cannot currently determine whether B and T cell activation was cell-autonomous due to exRNA recognition by their own innate sensors³¹, or requires prior activation of myeloid cells (**Figure S9**). Nevertheless, these results highlight that naked exRNAs can trigger systemic inflammation in vivo.

Minor Points:

1. The claim that the "main mechanism by which RI facilitates naked exRNA uptake is by stabilizing RNA" could be strengthened with titration studies to quantify what level of RNA is required to see effects in target cells.

We agree with the reviewer, but we would like to point out the original manuscript already contained titration experiments of RI both for the response of naked *E. coli* RNA in BMDCs (**new Figure 1E**) as well as for naked mRNA capture by this cell type (**new Figure 4D**). In addition, the original manuscript also contained titration experiments for the RNA in BMDCs (**new Figure 1D** and **3B-C**), and in vivo (**new Figure 6D**).

2. Is there a reason that an osteosarcoma cell line is used in Figure 5H/I? This seems quite biologically distinct from RNAs targeting antigen presenting cells (i.e. DC and macrophages which are used in most of the other models).

This study has a major focus on RNA immunology but also has some implications for mRNA therapeutics (the RI-dependent gymnotic uptake of mRNA is a quite unexpected results that, in our opinion, increases the interest of the manuscript for a broad audience).

All experiments oriented to understand naked RNA immunology were done in relevant cell models such as: BMDCs, Raw264.7 macrophages (a new addition of the revised version), THP-1 monocytes, THP-1 macrophages, and in vivo work.

The experiments using GFP and nanoLuc mRNAs are a two-edged sword because they serve both to demonstrate endosomal escape but also have biotechnological implications that could be extended to other cell types. For this reason, we decided to include data on U-2 OS and MCF-7 cells (**new Figure 4E-I**) in this part of the study, in addition to testing it also in BMDCs (**new Figure 4B-D**). We see this as a strength of the manuscript rather than a limitation.

3. Although it is possible that enhanced RNase activity in serum could protect from severe systemic inflammation (lines 205-206), there is no direct evidence to support this conclusion in the paper. It might be better to include this concept in the discussion rather than results section. How the balance of RNA sensing serves to benefit immune responses vs damage hosts is an important question that will require careful future work, requiring consideration of both the beneficial and detrimental effects of inflammation (lines 279-283).

We removed the following phrase from the results section: “*and suggest that extracellular RNases might have evolved to counteract this potentially life-threatening effect*”. We agree that the discussion is a more suitable place to include these far-reaching implications.

4. In Supp Fig 4B, what is the major population of cells that you gate out on FWD-SSC?

The heterogeneity in BMDC cultures differentiated with GM-CSF is known and has been studied in detail in several papers cited in the manuscript’s original version such as: Helft et al. (Immunity, 2015), Na et al. (Mol Cells 2016) and Erlich et al. (Nature Immunology, 2016). We also saw at least two subpopulations based on the FWD and SSC, which behaved differently in their poly(I:C) endocytosis rates, in agreement with the papers cited above.

Based on this complexity, and the reviewer’s request for more experimental replicates (see the following question), we decided to carry out all of the requested additional experiments in a more homogeneous system such as THP-1 cells (**new Figure 3F-G** and **S7A-B**). In addition, these cells are the relevant model in this chapter. For these reasons, we substituted original Supp Figure 4B with the new data mentioned above.

5. There are several places where imaging and flow experiments are not quantified over multiple replicates. This would help the reader have confidence in the robustness of the result ... ie Fig 2B, 2E, 5I, 6B, S6.

Thank you. Now the revised version of the manuscript contains multiple replicates for every key quantitative in vitro assay, including flow experiments (see our answer to your previous question). For the in vivo peritoneal data, the robustness of these results stems from quantifying immune populations across multiple concentrations of both naked *E. coli* and naked poly(I:C) (this last one included in the new version of the manuscript). In adherence to the 3R rule in animal experimentation, we opted not to use additional mice, as the existing experiments yielded clear results (see below data). We did increase the number of biological replicates for the i.v injection experiments, because these experiments were done at a single dose (see data below).

Referees' report, second round of review

Reviewer #2: In revision, the authors have increase the robustness of the study. They have added critical pieces of data to support their conclusions. The work will add important and though-provoking new data and new ideas for the scientific community.

Authors' response to the second round of review

None